**Influencing Factors of Gas-Particle Distribution of Oxygenated Organics in Urban**
**Atmosphere and Deviation from Equilibrium Partitioning: A Random Forest Model Study.**
Xinyu Wang[1], Nan Chen[2,3], Bo Zhu[2,3], Huan Yu[1,2] *
[1] School of Environmental Studies, China University of Geosciences, Wuhan, 430074, China
[2] Research Centre for Complex Air Pollution of Hubei Province, Wuhan 430074, China
[3] Hubei Ecological Environment Monitoring Center Station, Wuhan 430070, China
* Correspondence to: yuhuan@cug.edu.cn
**Abstract**
Gas-to-particle partitioning governs the fate of Oxygenated Organic Molecules (OOMs) and
the formation of organic aerosols. We employed a Chemical Ionization Mass Spectrometer equipped
with a Filter Inlet for Gases and AEROsol (FIGAERO-CIMS) to measure gas-particle distribution
of OOMs in a winter campaign in urban atmosphere. The observed gas to particle (G/P) ratios show
a narrower range than the equilibrium G/P ratios predicted from saturation mass concentration $C^*$
and organic aerosol content. The difference between observed and equilibrium G/P ratios could be
up to 10 orders of magnitude, depending on $C^*$ parameterization selection. Our random forest
models identified relative humidity (RH), aerosol liquid water content (LWC), temperature and
ozone as four influential factors driving the deviations of partitioning from equilibrium state.
Random forest models with satisfactory performance were developed to predict the observed G/P
ratios. Intrinsic molecule features far outweigh meteorological and chemical composition features
in the model's predictions. For a given OOM species, particle chemical composition features
including pH, RH, LWC, organic carbon, potassium and sulfate dominate over meteorological and
gaseous chemical composition features in predicting the G/P ratios. We identified positive or
negative effects, as well as the sensitive ranges, of these influential features using SHapley Additive
exPlanations (SHAP) analysis and curve fitting with a generalized additive model (GAM). Our
models found that temperature does not emerge as a significant factor influencing the observed G/P
ratios, suggesting that other factors, most likely associated with particle composition, inhibit the
gas/particle partitioning of OOMs in response to temperature change.

## 1. Introduction

Oxygenated organic molecules (OOMs) are ubiquitous in the atmosphere. They are key
constituents of organic aerosols (OA) and play a critical role in particle formation and growth (Yuan
et al., 2024). The distribution of an OOM between gas and particle phases not only reflects its
volatility or water solubility, but also governs its formation pathways, atmospheric transport and
deposition. Therefore, understanding the phase distribution of OOMs is essential for gaining
insights into their volatility, transformation processes and environmental impacts in the atmosphere.
Gas-to-Particle (G/P) ratios of OOMs measured by laboratory (e.g., ozonolysis products from
$\Delta^3$-Carene (Li et al., 2024a)) or field studies (e.g., in Hyytiälä forest, Finland (Lutz et al., 2019))
were sometimes used to derive saturation mass concentrations ($C^*$) or partitioning coefficients ($K_i$),
assuming that the observed G/P ratios represent an equilibrium partitioning state (Priestley et al.,
2024; Li et al., 2024a; Lutz et al., 2019; Stark et al., 2017). However, the G/P ratio of an OOM in
atmospheric conditions is influenced by not only intrinsic OOM physicochemical properties but
also external factors such as meteorological shifts (Hildebrandt et al., 2009), precursor oxidation
(Pankow, 1994; Seinfeld and Pankow, 2003), particle chemical composition, morphology and
particle-phase reactions (Jang et al., 2002; George et al., 2007). As a result, OOMs rarely achieve
equilibrium partitioning between the gas and particle phases (Roldin et al., 2014; Li et al., 2024b).
Gas/particle partitioning kinetics has been incorporated into many atmospheric aerosol models,
such as aerosol dynamics models (Liu et al., 2019; Zaveri et al., 2014) and kinetic multilayer models
(Fowler et al., 2018; Roldin et al., 2014), which accounted for molecular transfer rates, interphase
interactions, and environmental variability (Shiraiwa and Pöschl, 2021) in the gas-to-particle
transfer process. The development of these models has advanced our understanding of the
distribution and transport of organic compounds. However, existing theories and models often rely
on parameter simplifications or assumptions, and there is a lack of systematic studies examining the
factors influencing the phase distribution of OOMs under real atmospheric conditions. In recent
years, machine learning methods have been successfully applied for a variety of purposes including
compound identification (Franklin et al., 2022; Boiko et al., 2022), aerosol classification

(Christopoulos et al., 2018; Bland et al., 2022), precursor apportionment (Pande et al., 2022; Wang et al., 2021) and property prediction (Gong et al., 2022; Ruiz-Jimenez et al., 2021). Machine learning has been proven to be a powerful, data-driven approach capable of uncovering complex and nonlinear relationships between variables. (Lin et al., 2022; Zhu et al., 2019). Unlike physical or chemical models, machine learning does not rely on predefined assumptions or simplifications, which enables it to unveil previously unrecognized interactions.

In this work, we employed a Chemical Ionization Mass Spectrometer equipped with a Chemical Ionization Mass Spectrometer equipped with a Filter Inlet for Gases and AEROsol (FIGAERO-CIMS) in an urban location to measure the concentrations of OOMs in both the gas and particle phases. By building data-driven machine learning models with the G/P ratio as the target variable, we explored the influencing factors of gas-particle distribution of OOMs and examined the factors that contribute to the deviations from equilibrium gas/particle partitioning. This study offered new insights and provided the foundation for future studies on the atmospheric behavior of OOMs.

**2. Methodology**

**2.1 OOM measurement**

Hourly measurements of OOMs in both gas and particle phases was conducted during a winter campaign from December 5th, 2022, to January 8th, 2023, using an iodide-based FIGAERO-CIMS (Aerodyne Research Inc., USA) at a suburban site in Wuhan, a megacity in central China (114.6157°E, 30.4577°N). The site is located in the campus of China University of Geosciences, which is surrounded by residential and agricultural mixed area. The nearest urban center and industrial area are about 25 km west to the measurement site. Nearest highways and major roads lie about 2 km north and south of the site. The site is the only provincial supersite operated by local environmental authority for monitoring air quality in Wuhan and can thus be regarded as a receptor site influenced by wide ranges of emission sources from neighboring regions. We obtained valid data of 594 hours, during which meteorological parameters (e.g., relative humidity (RH) and temperature ($T$)), particulate chemical components (e.g., organic carbon (OC) and sulfate ions ($SO_4^{2-}$)), and gaseous components (e.g., sulfur dioxide ($SO_2$) and ozone ($O_3$)) were routinely

monitored. Detailed information about those routine measurement is shown in the supplementary
materials (Text S1).
The design of FIGAERO-CIMS for hourly OOMs measurement has been described by
previous studies (Zhao et al., 2024; Lopez-Hilfiker et al., 2014; Lee et al., 2014). Briefly, the
FIGAERO operated in a measurement cycle of 1 hour alternating between gas-phase and particle-
phase modes. During the gas-phase mode, ambient air was drawn at a flow rate of 2 L/min directly
into the ion-molecule reactor (IMR), where gaseous molecules were ionized and subsequently
detected as adduct ions with the reagent ion $I^-$. Simultaneously, another flow of ambient air was
pulled through a $PM_{2.5}$ cyclone (URG-2000-30EN, URG Corp.) and then a PTFE filter (2 μm
Zefluor, 25 mm, Pall Corp.), where particles smaller than 2.5 μm were collected. During the
subsequent particle-phase mode, the molecules on the PTFE filter underwent thermal desorption in
a heated ultrahigh-purity (UHP) nitrogen flow, which kept at room temperature for 2 minutes,
increased to 200 ℃ over 15 minutes, held at 200 ℃ for an additional 15 minutes to ensure the
desorption of the majority of OOMs (Lopez-Hilfiker et al., 2014) and then cooled to room
temperature within 4 minutes. The desorbed molecules were directed into a turbulent flow IMR. A
field blank sample was collected every 24 hours. According to our earlier investigation (Wang et al.
2024), The OOM measured with the FIGAERO-CIMS stands for only those polar and moderate-
volatility organic species being desorbed below 200°C and accounted for only 26 ± 8% of the total
OA (OC×1.4) measured with the thermal-optical method using the IMPROVE protocol.
**2.2 OOMs Identification and Selection**
OOMs were identified using a non-target strategy. Mass calibration was performed using ions
such as $NO_3^-$, $C_2F_3O_2^-$, $IC_2H_2O_2^-$, $IC_2F_3HO_2^-$, $IC_3F_5HO_2^-$, and $I_3^-$, covering a mass range from 62
to 381 m/z. The spectra peaks were iteratively fitted with multiple peaks using a custom peak shape
until the residual was reduced to less than 5 % (Lee et al., 2014; Stark et al., 2015). Subsequently,
the exact masses of these multiple peaks were matched with the most probable elemental formulas
within the ranges of $C_{1-30}H_{1-60}O_{0-20}N_{0-2}S_{0-2}X_{0-2}I_{0-1}^-$, where X stands for halogen atoms, with mass
errors smaller than 10 ppm (mass resolution of ~6000). Isotope distribution was inspected to match
with theoretical isotope pattern. Elemental ratio and double bond equivalent (DBE) limits of the
formulas were $0.3 \leq H/C \leq 3$, $N/C \leq 0.5$, $O/C \leq 3$, $S/C \leq 1$ and $0 \leq DBE \leq 20$ (Kind and Fiehn, 2007;
Lee et al., 2018; Kind and Fiehn, 2006).
In order to obtain reliable concentrations and thus G/P ratios, only those OOMs with a unit
mass peak area ratio of > 20 % and a sample-to-blank ratio of > 2 were included for further analysis.
This filtered out the OOMs with small concentrations in the atmosphere, as well as those extremely
high or low volatility OOMs that are predominantly in only one phase. Thermal desorption may
cause OOM decomposition in the particle phase. According to our earlier study on the same dataset
using a K-means clustering method (Wang et al., 2024), on average, 25.1% of particle-bound OOM
species number and 26.8% of OOM mass detected by the FIGAERO-CIMS could be attributed to
thermal decomposition fragments (Supplementary Materials Text S2). These fragments were
excluded from the gas/particle partitioning analysis. The overlap of non-fragment particle-bound
OOM species with those gas-phase OOM species resulted in 123 species, which were chosen as the
target species for subsequent partitioning analysis. Based on our previous work (Figure S1) (Wang
et al., 2024), these 123 OOM species were classified to 41 aromatic species (33.7%), 35
monoterpene-derived species (28.3%), 14 isoprene-derived species (11.4%), 11 aliphatic species
(8.7%), 10 biomass burning tracers (8.1%), 3 sulfur-containing species (2.4%) and 9 other unknown
species (7.3%). Notably, we cannot rule out the possibility that some of these 123 OOMs underwent
partial thermal decomposition in the particle phase to an unknown extent. This could lead to an
underestimate of their particle-phase concentrations and, in turn, bias the results toward higher G/P
ratios.
**2.3 Observed G/P ratios of OOMs**
The concentrations of an OOM species in gas phase and particle phase are calculated as:
$$C_g = \frac{signal_g}{S \times t_g \times Q_g} \times 1000 \qquad (1)$$

$$C_p = \frac{signal_p}{S \times t_p \times Q_p} \times 1000 \qquad (2)$$

where $C_g$ (ng m$^{-3}$) and $C_p$ (ng m$^{-3}$) are average concentrations of a species in gas phase and
particle phase, respectively, in a measurement interval (e.g., 1 hour in our campaign). $signal_g$ is
the integrated signal (unit: counts) of this species during the 21-minute gas-phase measurement time
$(t_g)$ in a measurement interval. $t_p$ is the particle sampling time (24 minutes) in a measurement
interval. $signal_p$ is the integrated signal of the particle-phase species during thermal desorption
(30 minutes) period. $Q_g$ and $Q_p$ are the sampling flow rates for the gas phase and particle phase,
respectively (Liter min[-1]). S is the sensitivity of the species (counts per ng). The observed G/P ratio
$(\frac{G}{P})_{obs}$ can be calculated as:
$$\left(\frac{G}{P}\right)_{obs} = \frac{C_g}{C_p} = \frac{signal_g \times t_p \times Q_p}{signal_p \times t_g \times Q_g} \qquad (3)$$

**2.4 Comparison with equilibrium G/P ratios**
According to modified Raoult's Law, the saturation ratio of an organic species in gas phase (i.e.
$\frac{C_g}{C^*}$) equals the mass fraction of the species in organic aerosol with mass concentration $C_{OA}$ ($i.e. \frac{C_p}{C_{OA}}$),
under the assumptions of equilibrium absorptive partitioning of the species over an ideal organic
solution and that the species has a molecular weight similar to that of the organic solution (Donahue
et al., 2009; Epstein et al., 2010). The equilibrium G/P ratio $(\frac{G}{P})_{eq}$ can thus be estimated from
saturated mass concentration $C^*$ and mass concentration of organic aerosol $C_{OA}$ ($C_{OA} = C_{OC} \times 1.4$)
using Eq. (4)
$$\left(\frac{G}{P}\right)_{eq} = \frac{C^*(T)}{C_{OA}} \qquad (4)$$

$C^*$ at 300 K of OOMs was calculated using 4 different parameterizations reported by Mohr et
al. (2019) , Peräkylä et al. (2020), Ren et al. (2022) and Priestley et al. (2024). Based on the
saturation concentrations of HOM modeled by Tröstl et al. (2016), Mohr et al. (2019) applied an
updated version of SIMPOL-type parameterization described by Donahue et al. (2011) to estimate
$C^*$ from the numbers of carbon, oxygen, and nitrogen atoms of an organic species ($n_C$, $n_O$, and $n_N$),
but emphasizing the increased importance of OOH groups. This parameterization likely produces
$C^*$ of pure compounds without considering the effect of particle matrix. Ren et al. (2022) obtained
$C^*$ of OOMs via calibrated $C^*$ vs. $T_{max}$ (thermal desorption temperature at which the maximum signal
intensity occurs) correlations in thermal desorption process. A similar parameterization formula
between $C^*$ and $n_C$, $n_O$, and $n_N$ was then derived using multivariate regression. Peräkylä et al.

(2020) parameterized the dependence of $C^*$ on $n_C$, $n_O$, $n_N$ and number of hydrogen atoms ($n_H$) by comparing steady-state gas-phase concentrations of α-pinene ozonolysis products with and without seed addition in a chamber. This parameterization predicts much smaller sensitivities of HOMs volatility to oxygen-containing functional groups than SIMPOL. The parameterization of Priestley et al. (2024) was based on measured gas and particle-phase concentrations, at an assumed equilibrium state, in residential wood-burning emissions. The $C^*$ of the products were obtained via Eq. (4) and a parameterization was obtained between $C^*$ and $n_C$, $n_O$, $n_N$ and $n_H$. The four $C^*$ parameterizations are listed in Text S3. A temperature correction was made based on Eqs. (5) and (6) to convert $C^*(300K)$ to $C^*(T)$ at observed temperatures (Epstein et al., 2010; Li et al., 2024a):

$$C^*(T) = C^*(300K) \times \exp\left(\frac{\Delta H_{vap}}{R}\left(\frac{1}{300K} - \frac{1}{T}\right)\right) \qquad (5)$$

$$\Delta H_{vap} = -11 \times \log_{10} C^*(300K) + 129 \qquad (6)$$

where $\Delta H_{vap}$ is the enthalpy of vaporization. $R$ is gas constant. $T$ is the observed temperature in every hour. $C^*(T)$ was then used in Eq. (4) to estimate equilibrium G/P ratios.

**2.5 Random forest model**

**2.5.1 Build random forest models**

Complex interactions and potentially non-linear dependences exist among OOM gas-particle partitioning, atmospheric chemistry, and environmental variables. We employed random forest models to investigate the influencing factors of gas-particle partitioning.

Our selection of influencing factors (i.e. features) is based on a comprehensive literature review. We categorized 30 features into four groups: (1) 9 molecule features of the OOMs: $n_C$, $n_O$, $n_N$, $n_H$, molecular weight (Mw), double bond equivalent (DBE), hydrogen to carbon atom ratios (H/C), oxygen to carbon atom ratios (O/C) and oxidation state of carbon (OSc). (2) 7 meteorological features: RH, $T$, wind speed (WS), wind direction (represented by sine and cosine functions to preserve the periodicity, denoted as WD_sin and WD_cos), ultraviolet-A (UV-A), ultraviolet-B (UV-B), $J_{HONO}$. (3) 4 gaseous composition features: $SO_2$ concentration, $O_3$ concentration, nitrogen dioxide ($NO_2$) concentration and ammonia ($NH_3$) concentration. (4) 10 particle composition features:

OC concentration, elemental carbon (EC) concentration, $SO_4^{2-}$ concentration, nitrate ions ($NO_3^-$)
concentration, chloride ions ($Cl^-$) concentration, ammonium ions ($NH_4^+$) concentration, $PM_{2.5}$
concentration, potassium ions ($K^+$) concentration, as well as aerosol-phase pH and liquid water
content (LWC). Calculation details of pH and LWC using ISORROPIA-II model (Fountoukis and
Nenes, 2007) are provided in Text S4. This feature selection scheme guarantees a balanced
representation of pertinent factors, while preserving the simplicity and predictive efficacy of the
models. Unlike neural networks and other machine learning algorithms, the random forest model
used in this study is an ensemble model made up of multiple decision trees. During training, each
tree splits using a randomly chosen subset of features. Because each tree uses different feature
subsets, this randomness in feature selection reduces the model's reliance on any single feature,
making it less likely to be severely impacted by multicollinearity. To further ensure model stability,
we also conducted five-fold cross-validation to confirm the robustness of the model.
First, we developed a multi-species model involving 123 OOM species to predict the $(\frac{G}{P})_{obs}$
of OOMs from molecule features, meteorological features, gas and particle composition features. A
total of 73062 $(\frac{G}{P})_{obs}$ values for 123 species with hourly resolution were collected in the winter
campaign. Outliers can indeed exacerbate modeling errors and potentially affect the model's
outcomes. Therefore, they should be removed (Leong et al., 2020). Outlier removal is described in
Text S5. The data used for modeling were randomly divided into training data (85% of the total) for
model training and test data (15% of the total) for evaluating model generalization.
Second, we selected six typical OOMs, including more volatile ($C_5H_8O_4$, $C_6H_{10}O_4$, $C_6H_5NO_3$,
$C_7H_7NO_3$, $C^*$ range: $10^{3.90}\sim10^{6.53}$ μg m$^{-3}$) and less volatile species ($C_{10}H_{16}O_4$, $C_{12}H_{21}NO_9$, $C^*$ range:
$10^{-4.73}\sim10^{1.18}$ μg m$^{-3}$) according to the $C^*$ parameterization of Mohr et al. (2019). $C_5H_8O_4$ (glutaric
acid (Lee et al., 2014; Reyes-Villegas et al., 2018)) and $C_6H_{10}O_4$ (adipic acid (Ye et al., 2021; Lee
et al., 2014)) are small dicarboxylic acids (C $\leqslant$ 6) typically formed through photochemical
degradation of reactions of alkenes, aldehydes, longer-chain acids (Kawamura and Sakaguchi, 1999)
or other low-oxygen organic compounds (Grosjean and Friedlander, 1980) in urban atmosphere
(Kawamura and Ikushima, 1993). $C_6H_5NO_3$ (Huang et al., 2019; Cai et al., 2022) and $C_7H_7NO_3$
(Huang et al., 2019; Cai et al., 2022) are nitrophenols either directly emitted from vehicle exhaust
(Tremp et al., 1993), coal and wood combustion (Huang et al., 2019), industrial processes (Harrison
et al., 2005) or being formed through the nitration of phenol in gas or liquid phase (Lüttke and
Levsen, 1997). $C_{10}H_{16}O_4$ is primarily derived the oxidation of monoterpenes (Ye et al., 2019;
Barreira et al., 2021). $C_{12}H_{21}NO_9$ is an organic nitrate from long-chain alkane oxidation under high-
$NO_x$ conditions (Wang and Ruiz, 2018).
Third, single-species models were tailored to predict the gas/particle partitioning behaviors of
these six individual OOMs under varying meteorological and gas-particle composition conditions.
We also built random forest models to investigate how $\left(\frac{G}{P}\right)_{obs}$ of the six OOMs deviate from $\left(\frac{G}{P}\right)_{eq}$
under varying meteorological conditions and gas/particle compositions. In this study, we did not
build random forest model to predict absolute gas or particle phase concentrations of OOMs, due to
their strong dependences on diverse emission sources from neighboring regions. We lack reliable
features for quantifying the variable strengths of unknown sources and atmospheric aging processes
during transport, which are key factors influencing the OOM concentrations.
**2.5.2 Model optimization, evaluation and feature importance analysis**
To optimize and evaluate the model's performance, we applied a combination of Grid Search
and Cross-Validation methods. First, we employed Grid Search to tune the hyperparameters of the
Random Forest model. The search space included the following hyperparameters: n_estimators (the
number of decision trees), max_depth (the maximum depth of each tree), and max_features (the
number of features considered for splitting at each node) and min_samples_split (the minimum
number of samples required to split an internal node). For each combination of hyperparameters,
we used 5-fold Cross-Validation on the training set with coefficient of determination ($R^2$) as the
primary metric to assess model performance and identify the best configuration. The specific
hyperparameter settings used in the Grid Search are provided in the supplementary materials, in
Table S1.
After selecting the optimal hyperparameters, we further evaluated the final model using 5-fold
cross-validation to assess the model's generalization ability and ensure it was not overfitted. In this
evaluation, both $R^2$ and Root Mean Square Error (RMSE) were used as metrics: $R^2$ indicates the

proportion of variance in the G/P ratio explained by the model. RMSE, on the other hand, quantifies the average prediction error and is calculated as the square root of the average squared differences between the predicted and actual values. The final model performance was determined by averaging the $R^2$ and RMSE values across the 5 validation sets. All model tuning and evaluation were conducted using Python (v.3.8).

To quantify the influence of each feature on the G/P ratio, we computed SHAP (SHapley Additive exPlanations) value of each feature for each sample (i.e., at each hour) using the SHAP package (v.0.40.0) in Python (v.3.8). A positive SHAP value indicates that the feature contributes positively to the G/P ratio, while a negative SHAP value means it has a negative contribution. The SHAP values versus feature values were then fitted with a generalized additive model (GAM) using the pygam package (v.0.8.0) to further identify the sensitive ranges where the changes of feature values significantly affect the SHAP values. For more details, please refer to Text S6. We utilized two-way Partial Dependence Plots (PDPs) (Chen et al., 2024a; Shi et al., 2023; Zhang et al., 2022) to analyze the joint effects of T and RH on the predicted G/P ratio. This analysis yielded a comprehensive understanding of how simultaneous changes of T and RH affect the observed G/P ratio, thereby unveiling the complex dynamics among these variables. For more details, please refer to the Text S7.

**3. Results and Discussion**

Despite the overall improvement in air quality in recent years, $PM_{2.5}$ episodes still occur frequently in December and January in most Chinese cities, contributing to the majority of $PM_{2.5}$ exceedance days of a year. During the winter observation period of this study, $PM_{2.5}$ concentrations ranged from 20 to 150 $\mu g\ m^{-3}$, spanning both clean and severe pollution conditions. Organic aerosol ($C_{OA} = C_{OC} \times 1.4$) comprised 10%–76% of $PM_{2.5}$, emerging as a critical bottleneck of eliminating $PM_{2.5}$ episodes. Time series of other criteria pollutants and key meteorological parameters are presented in Figure S2. The data collected during the observation period herein is considered representative of winter $PM_{2.5}$ pollution characteristics in Wuhan.

**3.1 Observed G/P ratios of OOMs and comparison with equilibrium partitioning**

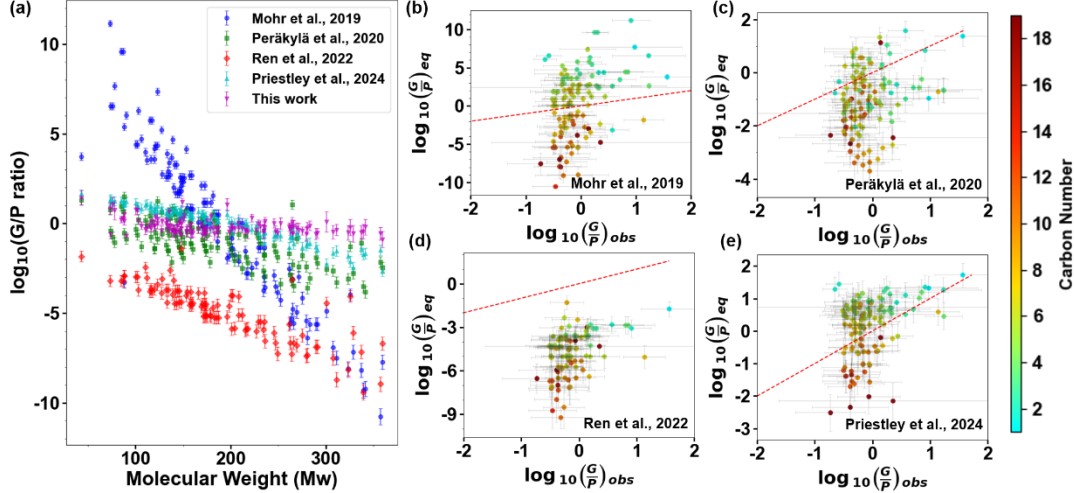

**Figure 1.** Comparison of $(\frac{G}{P})_{obs}$ of 123 OOMs with corresponding $(\frac{G}{P})_{eq}$ predicted by Eq. (4). $C^*$ was estimated from the parameterizations of Mohr et al. (2019), Peräkylä et al. (2020), Ren et al. (2022) and Priestley et al. (2024), respectively. Error bars of $(\frac{G}{P})_{obs}$ denote the range of G/P ratios observed under varying conditions for 594 samples (i.e. 594 hours). Error bars of $(\frac{G}{P})_{eq}$ denote the variations with temperature and $C_{OA}$. Color scales in (b-e) denote carbon number of OOM species. Dashed red lines in (b-e) denote a 1:1 correspondence.

As shown in Figure 1a, although G/P ratios generally decrease with increasing molecular weight, the observed G/P ratios $(\frac{G}{P})_{obs}$ show a narrower range ($10^{-1}\sim10^{1.5}$) than the equilibrium G/P ratios $(\frac{G}{P})_{eq}$ predicted from Eq. (4). The differences could be up to 10 orders of magnitude, depending on $C^*$ parameterization. Among all the methods, Mohr et al. (2019) predicts the steepest dependence of $(\frac{G}{P})_{eq}$ on Mw. Their $(\frac{G}{P})_{eq}$ are higher than $(\frac{G}{P})_{obs}$ for the OOMs with $n_C$ = 2-5 and lower than $(\frac{G}{P})_{obs}$ for the OOMs with $n_C$ > 8 (Figure 1b). It has been recognized by Kurtén et al. (2016) and subsequent publications that SIMPOL-derived parameterizations predict a too steep dependence of $C^*$ on Mw and oxygen content. Moreover, the parameterization of Mohr et al. (2019) likely produces $C^*$ of pure compounds. Without considering the effect of particle matrix, it may be

unrealistic to predict G/P ratios using their $C^*$ parameterization. On the basis of thermal desorption
temperature, Ren et al. (2022) predicts lower equilibrium G/P ratios than all other parameterizations
and our observation. The weakness of Ren et al. (2022) is thermal desorption may result in the
formation of decomposed fragments, which could be misidentified as OOM species. As a result, the
$T_{max}$ of OOM formulas tends to be overestimated and the $C^*$ tends to be underestimated in their
parameterization. Although Peräkylä et al. (2020) also predicted lower G/P ratios, their ratios are
much closer to our observation than Ren et al. (2022). Among all the predictions, the prediction
from Priestley et al. (2024) is most close to our observation. This is because their $C^*$
parameterization is based on the measured gas and particle-phase concentrations of OOMs in fresh
or aged residential wood-burning emissions. Their predicted G/P ratio is thus inherently consistent
with the observed G/P ratios in our study. This also highlights the risks of estimating volatility ($C^*$)
using the partitioning method, which is based on measuring equilibrium gas- and particle-phase
concentrations of OOMs. Two key issues arise: (1) OOMs may not achieve the assumed equilibrium
state in real atmospheric or chamber conditions, introducing substantial uncertainty into calculations
of $\left(\frac{G}{P}\right)_{eq}$; (2) The method fails for the compounds with extremely high or low volatility, as their gas-
or particle-phase concentrations often fall below the detection limit of mass spectrometers. These
limitations explain why the partitioning method typically reports a narrow volatility range (Voliotis
et al., 2021; Chen et al., 2024b).

306        In theory, no matter which $C^*$ parameterization is used in Eq. (4), the temporal variation of

$\left(\frac{G}{P}\right)_{eq}$ for an OOM species depends solely on $C_{OA}$ and temperature. Therefore, we are able to obtain
a normalized $\left(\frac{G}{P}\right)_{eq}$, which is independent of $C^*$ parameterization, by dividing the $\left(\frac{G}{P}\right)_{eq}$ of an
OOM by its maximum value. Diurnal variations of normalized $\left(\frac{G}{P}\right)_{eq}$ of $C_5H_8O_4$ and $C_7H_7NO_3$ are
shown in Figure 2a-2b and those of other four selected OOMs are shown in Figure S3. We found
similar diurnal variations for all six OOMs: a peak G/P ratio appeared in the afternoon, which is
attributed to higher temperature. In contrast, we observed different patterns of $\left(\frac{G}{P}\right)_{obs}$ diurnal
variations for the six OOM species during the campaign, as shown in Figure 2c-2h. This indicates
that the extent of deviation of actual gas/particle partitioning from equilibrium state fluctuates

randomly over time, driven by other unknown factors. In this study, we will first examine the influencing factors of gas-particle distribution of OOMs in urban atmosphere during the winter campaign (Section 3.2), followed by an investigation into the factors contributing to the discrepancies between observed and equilibrium G/P ratios (Section 3.3).

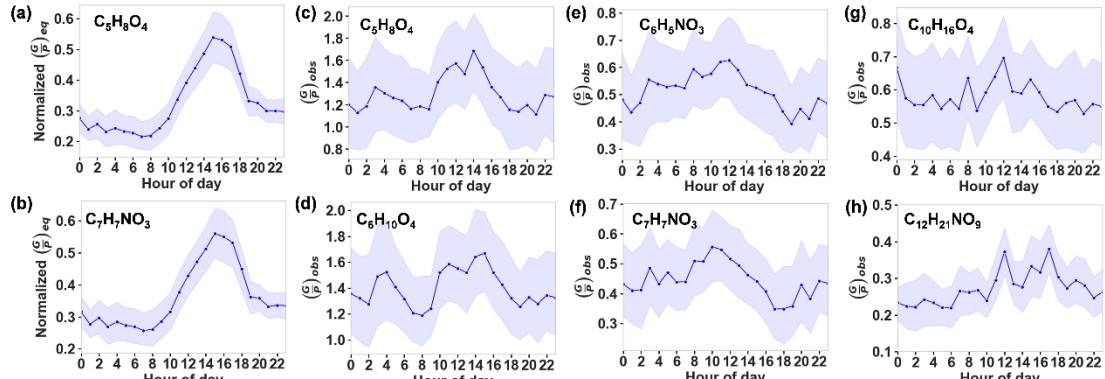

**Figure 2.** Diurnal variations of (a-b) normalized equilibrium G/P ratios for the selected species ($C_5H_8O_4$ and $C_7H_7NO_3$) and (c-h) observed G/P ratios during the campaign. Solid line denotes the average value and filled area denotes the 95% confidence intervals of the mean.

**3.2 Influencing Factors of the observed G/P ratios of OOMs**

**3.2.1 Multi-species model performance and key features**

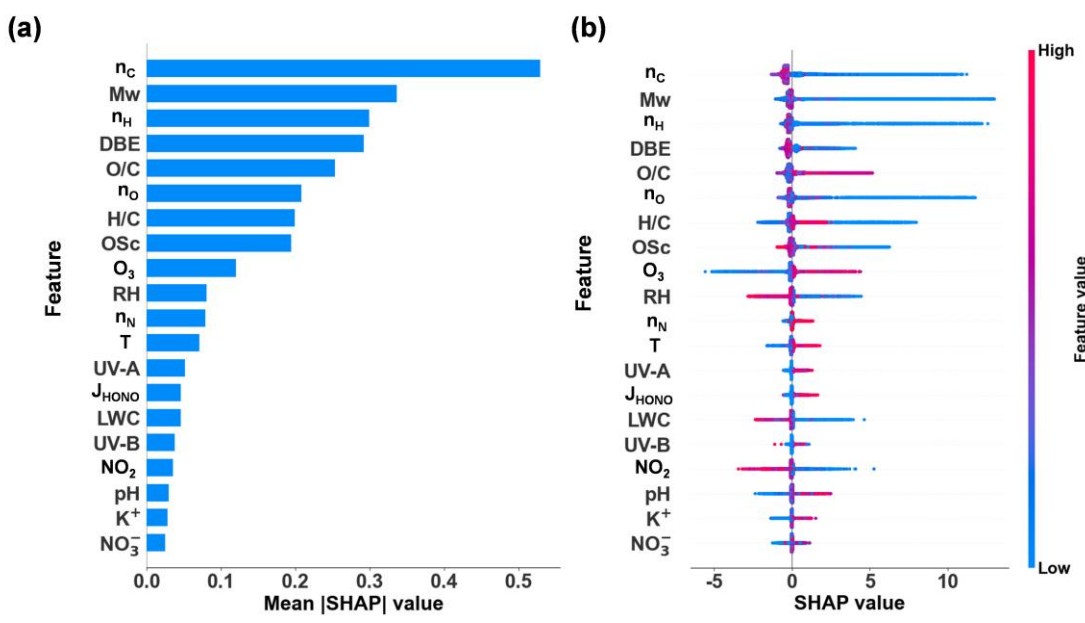

**Figure 3.** Multi-Species Model: (a) Feature importance based on the mean of absolute SHAP values

calculated for 594 samples (i.e. 594 hours) to predict the G/P ratio. (b) Distribution of SHAP values

in 594 samples for top 20 features.

The 5-fold cross-validation demonstrates that a predictive multi-species model with satisfactory generalization performance was developed, achieving $R^2 = 0.88 \pm 0.02$ and $RMSE = 1.76 \pm 0.13$ on the test set (Figure S4). Mean absolute SHAP values indicate the average importance of each feature in predicting the observed G/P ratios (Figure 3a). The model highlights that intrinsic molecule features, such as $n_C$, Mw, $n_H$, DBE, far outweigh meteorological and chemical composition features in the model's predictions. Of the nine molecular features, eight are ranked as highly important, with $n_N$ being comparatively less influential.

Figure 3b shows the SHAP value distribution for each feature. For molecule features, such as $n_C$, $Mw$, $n_H$ and $n_O$, high feature values are associated with negative SHAP, while low feature values are associated with positive SHAP. This suggests that large molecules with high $n_C$, Mw, $n_H$ and $n_O$, and consequently lower volatility, are more likely to partition into the particle phase, thereby reducing the G/P ratio.

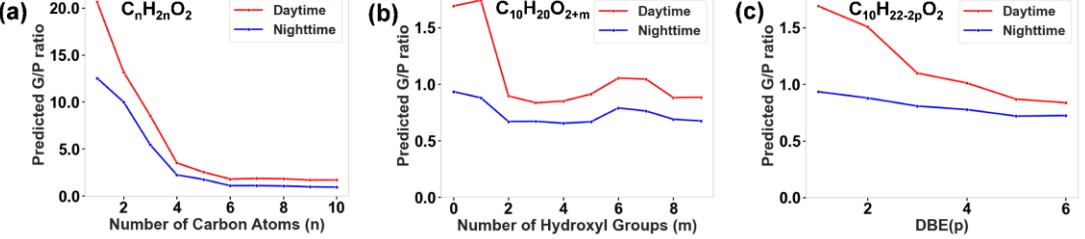

**Figure 4.** Predicted G/P ratios using the developed multi-species model for (a) Monocarboxylic acids as a function of the number of carbon atoms, (b) Modified 10-carbon monocarboxylic acids as a function of the number of additional hydroxyl groups and (c) Modified 10-carbon monocarboxylic acids as a function of DBE, under average daytime and nighttime environmental and gas/particle composition conditions.

However, the molecule features related to oxidation state and unsaturation degree did not show consistent effects on the observed G/P ratios. For example, OSc has a negative effect on the G/P ratios, whereas O/C has a positive effect. DBE has a negative effect on the G/P ratios, whereas H/C shows a mixed positive or negative effect. This is due to the fact that these features are dependent

variables as a function of $n_C$, $n_H$, $n_N$ and $n_O$. To isolate the effect of oxidation and unsaturation-related features, we utilized the trained random forest model to predict G/P ratios of modified $C_{10}$ monocarboxylic acid with varying number of hydroxyl group and DBE (Figure 4b and 4c). Other features in the model were fixed at average daytime or nighttime values observed during the campaign (see Table S2, S3). For comparison, the isolated effect of carbon atom number is also plotted (Figure 4a).

Figure 4 demonstrates that the number of carbon atoms exerts the most significant influence on the predicted G/P ratio, which decreases sharply as the carbon atom number increases from 1 to 4. Beyond this point, the ratio levels off. For modified 10-carbon monocarboxylic acids, G/P ratios are high when there is one or no hydroxyl group (Figure 4b). The G/P ratio levels off when the number of hydroxyl group exceeds 2. The G/P ratio decreases with increasing DBE value (Figure 4c). When DBE value exceeds 5, the G/P ratio change becomes minimal. In all the subplots, the G/P ratio during nighttime is consistently lower than that during daytime, which could be attributed to enhanced partitioning from gas to particles at lower nighttime temperature.

**3.2.2 Identification of key features and sensitive analysis in single-species models**

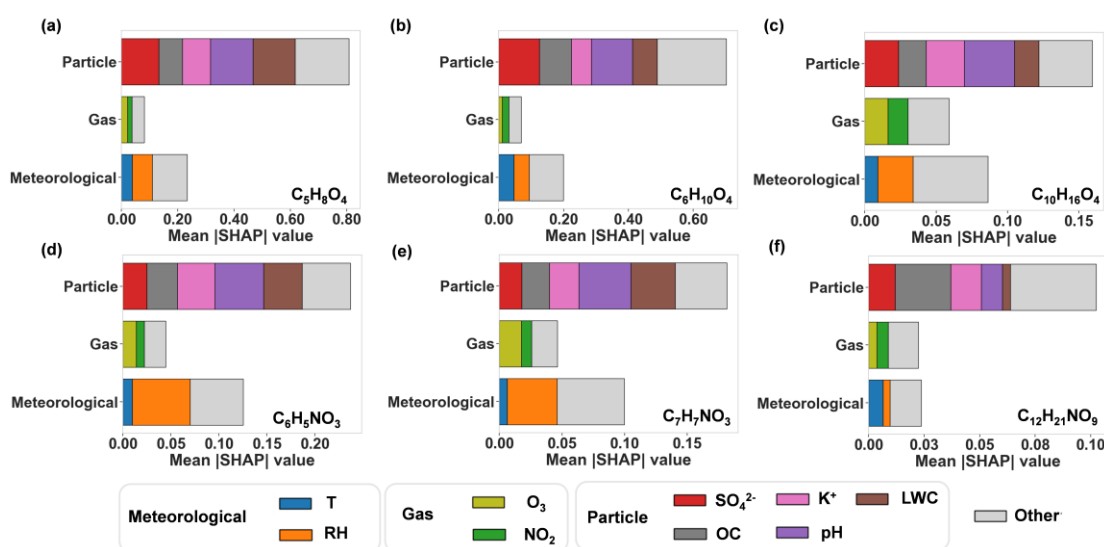

**Figure 5**. SHAP value analysis of three categories of features. Mean |SHAP| denotes the mean absolute SHAP values calculated for 594 samples (i.e. 594 hours): (a) glutaric acid ($C_5H_8O_4$), (b) adipic acid ($C_6H_{10}O_4$), (c) monoterpene oxidation products ($C_{10}H_{16}O_4$), (d, e) nitrophenol ($C_6H_5NO_3$

and $C_7H_7NO_3$), and (f) nitrated aliphatic acid ($C_{12}H_{21}NO_9$).
By excluding molecule features, single-species models focus on the prediction of observed
gas/particle partitioning behaviors of individual OOMs from meteorological and gas/particle
composition features. The evaluation results and optimal parameters of the six single-species models
on the test set are presented in Table S4. All models show acceptable generalization ability ($R^2$ =
0.51-0.88). For all six OOMs, particle composition features dominate over meteorological and
gaseous composition features in predicting the G/P ratios (Figure 5). Particle composition features
LWC, OC, $K^+$, $SO_4^{2-}$ and pH, as well as RH, consistently play important roles in influencing the G/P
ratios of these species. This is roughly in line with the correlation analysis between the features and
the observed G/P ratios of the selected 6 OOMs (Figure S5), which show that pH, RH, LWC, and
$SO_4^{2-}$ exhibited strong positive or negative correlations with the G/P ratios. Below, we (1) examined
the positive or negative effects of these features one by one (Figure 6a), and (2) identified the
sensitive ranges of these features by fitting SHAP values against feature values using a GAM (Figure

383    7).

pH is among the two most influential factors for the gas/particle partitioning of five species
($C_5H_8O_4$, $C_6H_{10}O_4$, $C_6H_5NO_3$, $C_7H_7NO_3$ and $C_{10}H_{16}O_4$) with a sensitive range of 3.5–4.5 (as
illustrated for $C_6H_{10}O_4$ in Figure 7a). Within this range, the contribution to the G/P ratio decreases
by 0.5 from pH 3.5 to 4.5. Beyond pH 4.5, the G/P ratio stabilizes at -0.1. An increase in pH results
in a pronounced decrease of the G/P ratio. This phenomenon can be attributed to the enhanced
partitioning of OOMs with acidic functional groups from gas to particles with elevated pH (Su et
al., 2020).
RH has a positive effect, ranking among the top 5 significant features, on the G/P ratios of three
OOMs $C_6H_5NO_3$, $C_7H_7NO_3$, and $C_{10}H_{16}O_4$ (Figure 6a). SHAP value is sensitive to RH across the
full RH range (20%-80%, illustrated by an example $C_6H_5NO_3$ in Figure 7b). LWC also has a
significant positive effect for $C_5H_8O_4$, $C_6H_{10}O_4$, $C_6H_5NO_3$, and $C_7H_7NO_3$. For example, in the case
of $C_5H_8O_4$, a sharp increase of 0.35 in the G/P ratio is observed within the LWC range below 20 μg
$m^{-3}$. Above 20 μg $m^{-3}$, the contribution to the G/P ratio stabilizes at 0.15 (Figure 7c). The underlying
mechanism of this behavior is unclear. One explanation is that the low RH and LWC in particles
may facilitate the reversible formation of oligomers (Shen et al., 2018) and suppress their hydrolysis
(Liu et al., 2012), thereby increasing the concentration of these OOMs in particle phase. It is also
possible that the thermal desorption and subsequent detection of particle-bound OOMs were
inhibited in aerosol particles with more moisture.

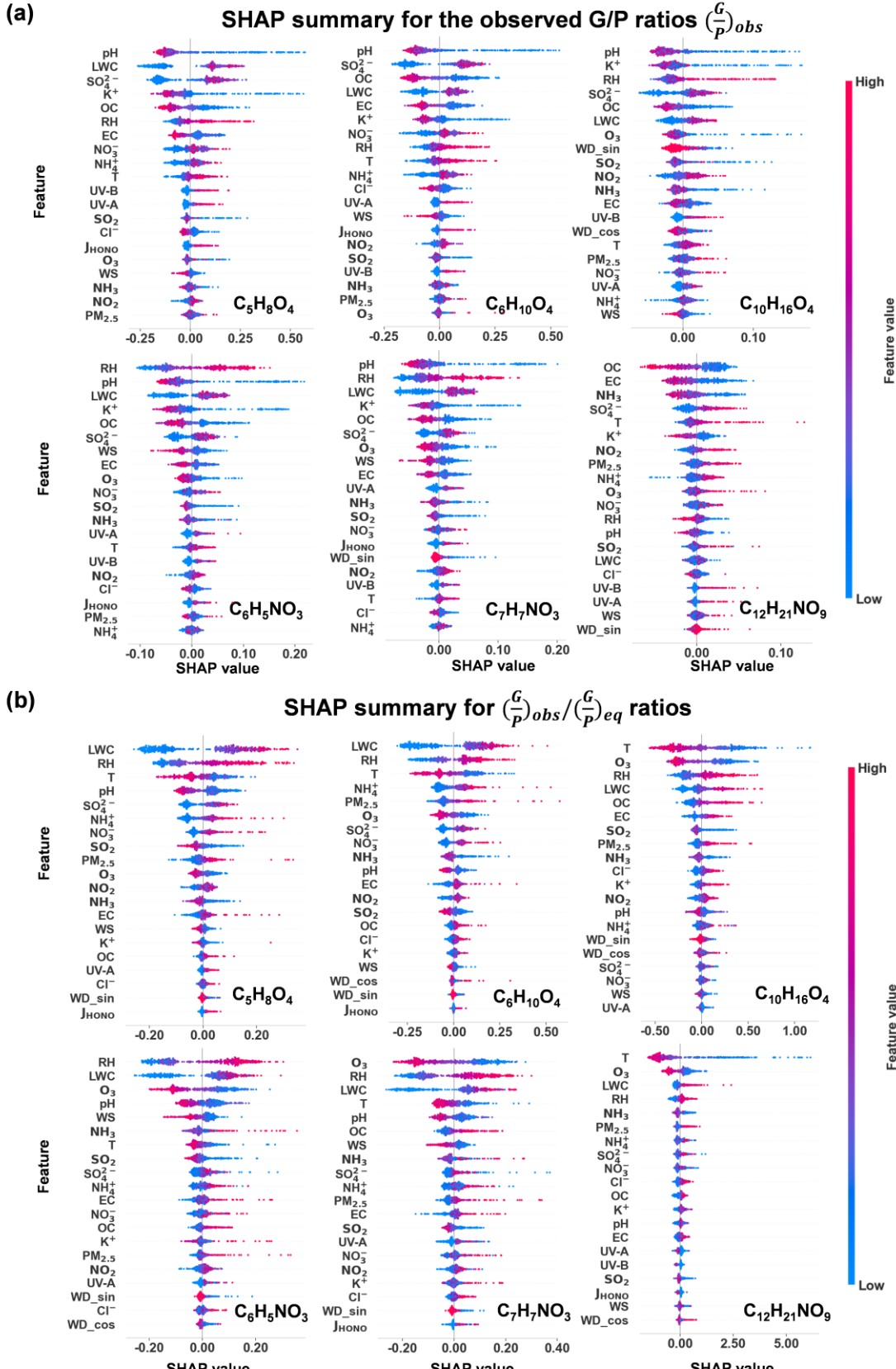


**Figure 6**. SHAP summary plots for feature importance based on the random forest model for glutaric

acid ($C_5H_8O_4$), adipic acid ($C_6H_{10}O_4$), monoterpene oxidation product ($C_{10}H_{16}O_4$), nitrophenol ($C_6H_5NO_3$ and $C_7H_7NO_3$), and nitrated aliphatic acid ($C_{12}H_{21}NO_9$). Features are prioritized in descending order based on their importance. (a) SHAP summary for the observed G/P ratios $(\frac{G}{P})_{obs}$.

(b) SHAP summary for $(\frac{G}{P})_{obs}/(\frac{G}{P})_{eq}$ ratios.

OC has a significant negative impact (i.e., rank among the top 5) on the G/P ratios of all six species, being consistent with Eq. (4), where the equilibrium G/P ratios are inversely proportional to $C_{OA}$. Taking $C_{12}H_{21}NO_9$ as example (Figure 7d), the SHAP values decrease monotonically with $C_{OA}$ by 0.08 in the entire $C_{OA}$ range (5-25 µg m$^{-3}$). For this compound, EC ranks as the second most influential factor, exerting a notable negative impact below 4 µg m$^{-3}$. A significant G/P decrease of 0.05 was observed in this range (Figure 7e).

$SO_4^{2-}$ has a positive effect (i.e., rank among the top 5) on the G/P ratios of $C_5H_8O_4$, $C_6H_{10}O_4$, $C_{10}H_{16}O_4$ and $C_{12}H_{21}NO_9$. For example, in the case of $C_6H_{10}O_4$, the G/P ratio rises rapidly by 0.30 with increasing $SO_4^{2-}$ concentrations below 6 µg m$^{-3}$ (Figure 7f). Above 6 µg m$^{-3}$, the contribution to the G/P ratio stabilizes at 0.1. This may be partly related to the fact that $SO_4^{2-}$ is a highly hydrophilic component (Thaunay et al., 2015), which makes its effect similar to that of LWC. In addition, an increase of sulfate in aerosols is often associated with enhanced acidity and a decrease in pH (Zhang et al., 2007), which drives OOM from particle to gas phase as we explained above.

$K^+$ has a negative effect on the G/P ratios of $C_5H_8O_4$, $C_{10}H_{16}O_4$, $C_6H_5NO_3$ and $C_7H_7NO_3$. Taking $C_{10}H_{16}O_4$ as example, the G/P ratio decreases rapidly by 0.15 with $K^+$ in the concentration range of below 1 µg m$^{-3}$. Above 1 µg m$^{-3}$, its contribution to the G/P ratio stabilizes at -0.03 (Figure 7g). $K^+$ is considered as a tracer of biomass burning. The increase of $K^+$ is generally associated with higher pollution levels and higher OC concentrations in the study region (Zhao et al., 2024). The effect of $K^+$ on the G/P ratio is thus similar to that of OC.

In general, temperature is supposed to be an important influential factor of G/P ratio, because saturation vapor pressure of OOMs increases with temperature. Temperature ranged from -1.6 °C to 14.9 °C during the winter campaign. Although temperature increase tends to elevate the G/P ratios as expected (Figure 6a), the models show that temperature did not rank as important feature for 5

out of the 6 OOM species. We evaluated the effect of temperature on G/P ratios using two-way
partial dependence plots (Figure S6). G/P ratio is sensitive to temperature change only for two
dicarboxylic acids ($C_5H_8O_4$ and $C_6H_{10}O_4$, Figure S6a-S6b) and for $C_{12}H_{21}NO_9$ in a narrow
temperature range of 10-13 °C (Figure S6f and Figure 7h). The G/P ratios of $C_6H_5NO_3$, $C_7H_7NO_3$
and $C_{10}H_{16}O_4$ are not sensitive to temperature across most of the RH range. This behavior may be
attributed to the aerosol coating of inorganic salts and other aerosol components that hinder the rapid
equilibrium partitioning of OOMs when temperature changes. In addition, the influence of
temperature may be obscured due to the dominant effect of particle composition features (e.g., LWC,
pH, OC, $SO_4^{2-}$, and $K^+$) as discussed above.

440       As shown in Figure 6a, wind speed and direction rank relatively low in terms of feature

importance for the six OOMs. This suggests that while wind direction and speed might influence
the source areas of OOMs, they have a minimal impact on the G/P ratios of OOMs.

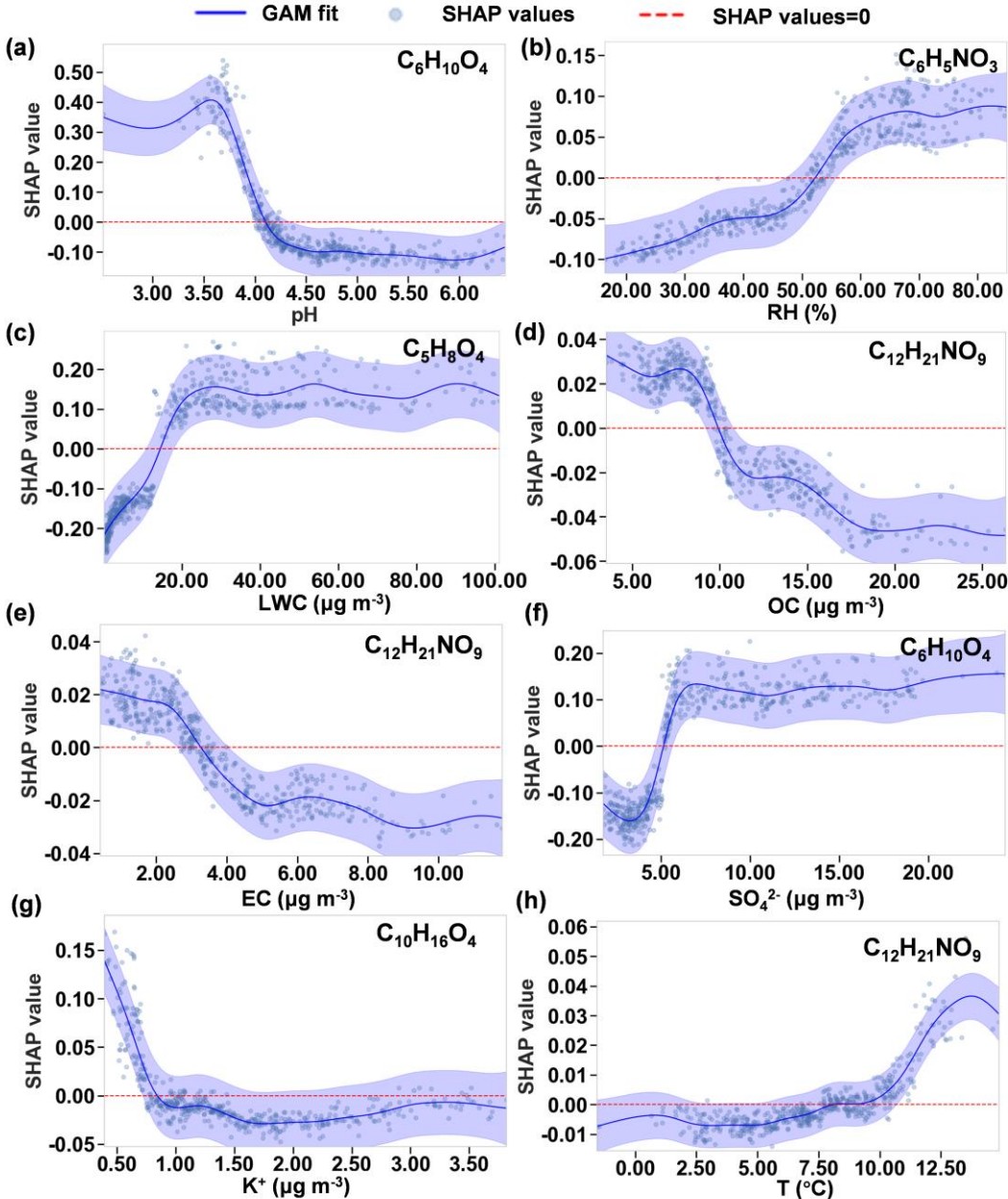


**Figure 7**. Curve fitting of SHAP values versus features using a GAM, illustrating the sensitive ranges where the changes of feature values significantly affect the SHAP values. Only the most affected OOM species by the eight features are shown. (a) pH for $C_6H_{10}O_4$. (b) RH for $C_6H_5NO_3$. (c) LWC for $C_5H_8O_4$. (d) OC for $C_{12}H_{21}NO_9$. (e) EC for $C_{12}H_{21}NO_9$. (f) SO4$^{2-}$ for $C_6H_{10}O_4$. (g) K$^+$ for $C_{10}H_{16}O_4$. (h) Temperature for $C_{12}H_{21}NO_9$. Blue line denotes the GAM fit. Shaded area indicates 95% confidence interval. Dots are the SHAP values for 594 samples (i.e. 594 hours). Red dashed line denotes SHAP value of 0.

**3.3 Identifying key factors driving the deviations of gas/particle partitioning from equilibrium state**

To investigate the deviations of observed gas/particle partitioning from equilibrium state, we first calculate the ratios of $(\frac{G}{P})_{obs}$ over normalized $(\frac{G}{P})_{eq}$ in every hour for the selected six OOM species. Normalized $(\frac{G}{P})_{eq}$ was used here in order to offset the effect of the $C^*$ parameterization selection. We then developed new random forest models to investigate the effects of meteorological and gas/particle composition features on the $(\frac{G}{P})_{obs}/(\frac{G}{P})_{eq}$ ratios. All the models show acceptable generalization performance ($R^2 = 0.52\text{-}0.83$) (Table S5) on the test set.

Figure 6b presents the SHAP analysis results for the $(\frac{G}{P})_{obs}/(\frac{G}{P})_{eq}$ ratios of the six OOMs. The models identify RH, LWC, $O_3$ and temperature as four influential factors driving the deviations from equilibrium partitioning. Positive correlations are observed between the SHAP values of $(\frac{G}{P})_{obs}/(\frac{G}{P})_{eq}$ and RH and LWC for all six compounds. This indicates that RH and LWC have stronger positive effect on $(\frac{G}{P})_{obs}$ than their effect on $(\frac{G}{P})_{eq}$, which should be negligible according to Eq. (4). Temperature is shown to be a negative factor driving the deviation from equilibrium partitioning, suggesting that temperature has a stronger influence on $(\frac{G}{P})_{eq}$ than $(\frac{G}{P})_{obs}$. This is consistent with our earlier result that $(\frac{G}{P})_{obs}$ is not sensitive to temperature. Surprisingly, $O_3$ is identified as an important influential factor with negative effect, particularly for the three nitrophenols and monoterpene oxidation product $C_{10}H_{16}O_4$. Since $O_3$ is not expected to change $(\frac{G}{P})_{eq}$, the negative impact of $O_3$ on $(\frac{G}{P})_{obs}/(\frac{G}{P})_{eq}$ ratio could be explained by the speculation (Kaur Kohli et al., 2023) that high $O_3$ concentrations are likely to deplete gas-phase OOMs at a faster rate than particle-phase OOMs, thereby reducing $(\frac{G}{P})_{obs}$.

**Conclusions**

We measured the G/P ratios of OOM species using a FIGAERO-CIMS in urban atmosphere in a winter campaign. The observed G/P ratios show a narrower range than the equilibrium G/P ratios

predicted from $C^*$ and $C_{OA}$. The difference between observed and equilibrium G/P ratios could be
up to 10 orders of magnitude, depending on $C^*$ parameterization. Our observed G/P ratio is
inherently closer to the equilibrium G/P ratios predicted from the $C^*$ parameterization by Priestley
et al., which was derived from measured G/P ratios in wood-burning emissions. Our random forest
models identified RH, LWC, $O_3$ and temperature as four influential factors driving the deviations of
gas/particle partitioning from equilibrium state.
Random forest models with satisfactory performance were developed to predict observed G/P
ratios. Intrinsic molecule features, such as $n_C$, Mw, $n_H$, DBE, far outweigh meteorological and
chemical composition features in the model's predictions. Large molecules with high $n_C$, Mw, $n_H$
and $n_O$, and consequently lower volatility, are more likely to partition into the particle phase, thereby
reducing the G/P ratio. As dependent variables, oxidation state and unsaturation do not show
consistently positive or negative effects on the observed G/P ratios. If other variables are fixed, the
model predicts that G/P ratios generally decrease with the addition of oxygen atom and DBE.
Particle composition features dominate over meteorological and gaseous composition features
in predicting the G/P ratio of a given OOM species. Among those particle features, pH, RH, LWC,
OC, $K^+$ and $SO_4^{2-}$ consistently play important roles in influencing the G/P ratios of the six selected
OOM species, showing either positive or negative effect. We also identified the sensitive ranges
where the changes of these features significantly affect the SHAP values and provided valuable
insights for future research in atmospheric chemistry. It is surprising that temperature does not
emerge as an important factor influencing the G/P ratios for five out of the six selected OOM species.
Our model suggests that other factors, most likely associated with the particle composition, inhibit
the gas/particle partitioning of OOMs in response to temperature change.
At last, the random forest models developed in this study have certain limitations. (1) Aerosol
particle coating may serve as an inhibitory factor of gas/particle partitioning. However, the mixing
state and morphology of aerosol particles were not considered in the model due to the challenges in
quantifying these features with high resolution. (2) The OOMs with extremely high or low volatility
might be underrepresented in this study, because their gas- or particle-phase concentrations often
fall below the limit of quantification of FIGAERO-CIMS. (3) Isomers were not differentiated in the

measurement of FIGAERO-CIMS in this study. The observed G/P ratio was contributed by isomers sharing the same chemical formula. The machine learning model built in this study did not account for the effect of isomerization on gas-particle distribution of OOMs. (4) The model was based solely on the data collected during the winter season and for specific groups of OOM species present in urban atmosphere. To enhance the robustness of the gas-to-particle partitioning model, future data collection under a broader range of atmospheric conditions is recommended.

**Data availability**

The data used in this article are available in the public data repository Zenodo https://doi.org/10.5281/zenodo.15428774.

**Author contributions**

HY designed the experiment. XW, BZ and NC contributed to data collection. XW and HY analyzed the data and wrote the manuscript.

**Competing interests**

The contact author has declared that none of the authors has any competing interests.

**Financial support**

This research was supported by the National Key Research and Development Program of China (2023YFC3709801), the National Natural Science Foundation of China (42175131) and the Fundamental Research Funds (No. G1323523063) for the Central Universities, China University of Geosciences (Wuhan).

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
