# Peer review of "Influencing Factors of Gas-Particle Distribution of Oxygenated Organics in Urban"

_EGUsphere, 2025_

## Author Comment (AC1)

**A point-to-point response and relevant changes made in the revised manuscript**

**Ms. Ref. No.: egusphere-2025-229**

**Title: Investigating Influencing Factors of Gas-Particle Distribution of Oxygenated Organic Molecules in Urban Atmosphere and Its Deviation from Equilibrium Partitioning Using Random Forest Model.**

**Anonymous Referee #1**

*Wang et al. measured both gaseous and particle-phase OOMs using FIGAERO-CIMS during a winter campaign in Wuhan. They derived gas-to-particle ratios (G/P) using measured FIGAERO-CIMS signals and predicted equilibrium G/P based on predicated OOM volatility. They further applied machine learning methods to revealed key factors that are associated with G/P. The manuscript aligns with the scope of Atmospheric Chemistry and Physics. However, clarification of the principal findings is recommended prior to publication. Specific comments are given below:*

**Response:**

We sincerely thank the reviewer for their valuable comments and suggestions, which will significantly improve the quality of our manuscript.

1. *The machine learning analysis provides intriguing insights into G/P influencing factors. However, given potential limitations in the representativeness of the dataset and the ML methodology, the mechanistic interpretation of identified factors may not be entirely clear. I recommend expanded discussion in Section 3.2.2 to include detailed analysis of at least one parameter (e.g., temperature).*

**Response:**

By itself, the machine learning methodology cannot uncover the fundamental mechanisms through which the factors influence G/P ratio. In Section 3.2.2, we (1) examined the positive or negative effects and the sensitive ranges of key features like RH, LWC, OC, $K^+$, $SO_4^{2-}$ and pH identified by the ML model and (2) explained why ambient temperature did not rank as important feature for most of the OOM species. Prior publications were provided to support our interpretation.

We expanded our discussion of the impact of temperature, RH and LWC. In lines 398-402,

"One explanation is that the low RH and LWC in particles may facilitate the reversible formation of oligomers (Shen et al. (2018) and suppress their hydrolysis (Liu et al., 2012), thereby increasing the concentration of these OOMs in particle phase. It is also possible that the thermal desorption and subsequent detection of particle-bound OOMs were inhibited in aerosol particles with more moisture."

In lines 428-440,

"In general, temperature is supposed to be an important influential factor of G/P ratio, because saturation vapor pressure of OOMs increases with temperature. Temperature ranged from -

1.6 °C to 14.9 °C during the winter campaign. Although temperature increase tends to elevate the G/P ratios as expected (Figure 6a), the models show that temperature did not rank as important feature for 5 out of the 6 OOM species. We evaluated the effect of temperature on G/P ratios using two-way partial dependence plots (Figure S6). G/P ratio is sensitive to temperature change only for two dicarboxylic acids ($C_5H_8O_4$ and $C_6H_{10}O_4$, Figure S6a-S6b) and for $C_{12}H_{21}NO_9$ in a narrow temperature range of 10-13 °C (Figure S6f and Figure 7h). The G/P ratios of $C_6H_5NO_3$, $C_7H_7NO_3$ and $C_{10}H_{16}O_4$ are not sensitive to temperature across most of the RH range. This behavior may be attributed to the aerosol coating of inorganic salts and other aerosol components that hinder the rapid equilibrium partitioning of OOMs when temperature changes. In addition, the influence of temperature may be obscured due to the dominant effect of particle composition features (e.g., LWC, pH, OC, $SO_4^{2-}$, and $K^+$) as discussed above."

2. *Please explain Eq. 4 with an emphasis on its underlying assumptions that are possibly violated in the real atmosphere. This clarification would aid the discussion on the influencing factors of the ratio of (G/P)$_{obs}$ to (G/P)$_{eq}$.*

**Response:**

Equation 4 is primarily based on Raoult's Law. In line 143-150, we explain the derivation of Eq 4 and underlying assumptions.

"According to modified Raoult's Law, the saturation ratio of an organic species in gas phase (i.e. $\frac{C_g}{C^*}$) equals the mass fraction of the species in organic aerosol with mass concentration $C_{OA}$ ($i.e. \frac{C_p}{C_{OA}}$), under the assumptions of equilibrium absorptive partitioning of the species over an ideal organic solution and that the species has a molecular weight similar to that of the organic solution (Donahue et al., 2009; Epstein et al., 2010). The equilibrium G/P ratio $(\frac{G}{P})_{eq}$ can thus be estimated from saturated mass concentration $C^*$ and mass concentration of organic aerosol $C_{OA}$ ($C_{OA} = C_{OC} \times 1.4$) using Eq. (4)

$$(\frac{G}{P})_{eq} = \frac{C^*(T)}{C_{OA}} \qquad\qquad (4)\text{ ''}$$

3. *Lines 179-181, Page 7. Please specify the data partitioning strategy for training and test sets and the measures to prevent model overfitting.*

**Response:**

Thank you for the reviewer's suggestion. Regarding the data partitioning strategy for the training and test sets, we performed a random split. In lines 207-208, we revised the text to:

"The data used for modeling were randomly divided into training data (85% of the total) for model training and test data (15% of the total) for evaluating model generalization."

Unlike gradient boosting regression and neural networks, the random forest algorithm trains multiple decision trees using bootstrapped subsets of the training data and random subsets of features, which inherently provides better resistance to overfitting (Amaratunga et al., 2008).

To further prevent model overfitting and enhance its generalization ability, we implemented the following 3 measures:

 1) Cross-validation: We employed 5-fold cross-validation on the training set to assess the model's generalization ability on different subsets of data. We revised the text in lines 242-243 to:

"After selecting the optimal hyperparameters, we further evaluated the final model using 5-fold cross-validation to assess the model's generalization ability and ensure it was not overfitted."

2) Restricting parameter ranges: In the Grid Search method, we restricted the parameter ranges for n_estimators, max_depth, max_features, and min_samples_split to prevent overfitting. In lines 239-241, we added the following sentence:

"The specific hyperparameter settings used in the Grid Search are provided in the supplementary materials, in Table S1."

In the supplementary materials, we have added Table S1:

Table S1. Hyperparameters for grid search in random forest model optimization

| Hyperparameter | Values |
|---|---|
| n_estimators | 50, 100, 150, 200 |
| max_depth | 10, 20, 30, None |
| min_samples_split | 2, 5, 10 |
| min_samples_leaf | 1, 2, 4 |
| max_features | sqrt, log2 |

3) Evaluation of Model Generalization Using the Test Set: The test set, which was not involved in the training process, provides a reliable assessment of the model's generalization ability, helping to prevent overfitting. The $R^2$ evaluations presented in Table S4 and Table S5 of the supplementary materials are based on the test set results, demonstrating the model's satisfactory generalization performance.

In lines 330-332, we revised the sentence to:

"The 5-fold cross-validation demonstrates that a predictive multi-species model with satisfactory generalization performance was developed, achieving $R^2$=0.88 ± 0.02 and RMSE = 1.76 ± 0.13 on the test set (Figure S4)."

In lines 374-376, we revised the sentence to:

"The evaluation results and optimal parameters of the six single-species models on the test set are presented in Table S4. All models show acceptable generalization ability ($R^2 = 0.51$-0.88)."

In lines 458-459, we revised the sentence to:

"All the models show acceptable generalization performance ($R^2 = 0.52$-0.83) (Table S5) on the test set."

4. *Fig. 1 and its relevant discussion. The method in Ren et al. (2022) provided the equilibrium*

pressure ($C^{eq}$) rather than the saturation pressure ($C^*$), as the formula was fit to atmospheric aerosols (a mixture of many OOMs). Using $C^{eq}$ instead of $C^*$ in Eq. 4 may introduce systematic biases. Could this be the reason for the observed discrepancy between $(G/P)_{obs}$ and $(G/P)_{eq}$ in Figs 1a and 1d?

**Response:**

In Ren et al. (2022), the authors first obtained a calibration curve for $C^*$ and $T_{max}$ using a series of polyethylene glycol standards with known saturation vapor pressures. Then they measured $T_{max}$ values of a number of OOMs in atmospheric aerosols and converted the $T_{max}$ values into corresponding $C^*$ of the OOMs. Therefore, Ren et al. indeed provided the saturation concentration ($C^*$) values of OOMs with known formulas, although the OOMs were obtained from ambient aerosols in their experiment.

5. *Previous studies (e.g., Voliotis et al., 2021; Chen et al., 2024) have reported a narrow volatility range of OOMs retrieved using the partitioning method. It is not surprising to see a huge difference between the volatility obtained using different methods. Would it be possible to expand discussion on this finding?*

**Response:**

Thank you for providing the relevant literature. We cited them and expanded the discussion in lines 295-306 as follows:

"Among all the predictions, the prediction from Priestley et al. (2024) is most close to our observation. This is because their $C^*$ parameterization is based on the measured gas and particle-phase concentrations of OOMs in fresh or aged residential wood-burning emissions. Their predicted G/P ratio is thus inherently consistent with the observed G/P ratios in our study.

This also highlights the risks of estimating volatility ($C^*$) using the partitioning method, which is based on measuring equilibrium gas- and particle-phase concentrations of OOMs. Two key issues arise: (1) OOMs may not achieve the assumed equilibrium state in real atmospheric or chamber conditions, introducing substantial uncertainty into calculations of $\left(\frac{G}{P}\right)_{eq}$; (2) The method fails for the compounds with extremely high or low volatility, as their gas- or particle-phase concentrations often fall below the detection limit of mass spectrometers. These limitations explain why the partitioning method typically reports a narrow volatility range (Voliotis et al., 2021; Chen et al., 2024)."

6. *As noted by the authors, atmospheric OOMs may not reach equilibrium between the gas and particle phases (e.g., Li et al., 2024). Could machine learning features capture this non-equilibrium effects?*

**Response:**

Yes, machine learning features could capture the non-equilibrium effects. This was discussed in Section 3.3 "Identifying key factors driving the deviations of gas/particle partitioning from equilibrium state". The ML model identified RH, LWC, $O_3$ and temperature as four features that lead to non-equilibrium partitioning.

In lines 44-45, we add the citation Li et al., 2024:

"As a result, OOMs rarely achieve equilibrium partitioning between the gas and particle phases (Roldin et al., 2014; Li et al., 2024)."

**References**

Amaratunga, D., Cabrera, J., and Lee, Y.-S.: Enriched random forests, Bioinformatics, 24, 2010-2014, http://doi.org/10.1093/bioinformatics/btn356, 2008.

Chen, W., Hu, W., Tao, Z., Cai, Y., Cai, M., Zhu, M., Ye, Y., Zhou, H., Jiang, H., Li, J., Song, W., Zhou, J., Huang, S., Yuan, B., Shao, M., Feng, Q., Li, Y., Isaacman-VanWertz, G., Stark, H., Day, D. A., Campuzano-Jost, P., Jimenez, J. L., and Wang, X.: Quantitative Characterization of the Volatility Distribution of Organic Aerosols in a Polluted Urban Area: Intercomparison Between Thermodenuder and Molecular Measurements, J. Geophys. Res. Atmos., 129, e2023JD040284, https://doi.org/10.1029/2023JD040284, 2024.

Donahue, N. M., Robinson, A. L., and Pandis, S. N.: Atmospheric organic particulate matter: From smoke to secondary organic aerosol, Atmospheric Environment, 43, 94-106, https://doi.org/10.1016/j.atmosenv.2008.09.055, 2009.

Epstein, S. A., Riipinen, I., and Donahue, N. M.: A Semiempirical Correlation between Enthalpy of Vaporization and Saturation Concentration for Organic Aerosol, Environ. Sci. Technol., 44, 743-748, https://doi.org/10.1021/es902497z, 2010.

Li, Y., Cai, R., Yin, R., Li, X., Yuan, Y., An, Z., Guo, J., Stolzenburg, D., Kulmala, M., and Jiang, J.: A kinetic partitioning method for simulating the condensation mass flux of organic vapors in a wide volatility range, J. Aerosol Sci., 180, 106400, https://doi.org/10.1016/j.jaerosci.2024.106400, 2024.

Liu, S., E., S. J., Chen, S., Naruki, H., A., Z. R., and and Russell, L. M.: Hydrolysis of Organonitrate Functional Groups in Aerosol Particles, Aerosol Sci. Technol., 46, 1359-1369, http://doi.org/10.1080/02786826.2012.716175, 2012.

Priestley, M., Kong, X., Pei, X., Pathak, R. K., Davidsson, K., Pettersson, J. B. C., and Hallquist, M.: Volatility Measurements of Oxygenated Volatile Organics from Fresh and Aged Residential Wood Burning Emissions, ACS Earth Space Chem., 8, 159-173, https://doi.org/10.1021/acsearthspacechem.3c00066, 2024.

Roldin, P., Eriksson, A. C., Nordin, E. Z., Hermansson, E., Mogensen, D., Rusanen, A., Boy, M., Swietlicki, E., Svenningsson, B., Zelenyuk, A., and Pagels, J.: Modelling non-equilibrium secondary organic aerosol formation and evaporation with the aerosol dynamics, gas- and particle-phase chemistry kinetic multilayer model ADCHAM, Atmos. Chem. Phys., 14, 7953-7993, https://doi.org/10.5194/acp-14-7953-2014, 2014.

Shen, H., Chen, Z., Li, H., Qian, X., Qin, X., and Shi, W.: Gas-Particle Partitioning of Carbonyl Compounds in the Ambient Atmosphere, Environ. Sci. Technol., 52, 10997-11006, http://doi.org/10.1021/acs.est.8b01882, 2018.

Voliotis, A., Wang, Y., Shao, Y., Du, M., Bannan, T. J., Percival, C. J., Pandis, S. N., Alfarra, M. R., and

McFiggans, G.: Exploring the composition and volatility of secondary organic aerosols in mixed anthropogenic and biogenic precursor systems, Atmos. Chem. Phys., 21, 14251-14273, https://doi.org/10.5194/acp-21-14251-2021, 2021.

---

## Author Comment (AC2)

**A point-to-point response and relevant changes made in the revised manuscript**

**Ms. Ref. No.: egusphere-2025-229**

**Title: Investigating Influencing Factors of Gas-Particle Distribution of Oxygenated Organic Molecules in Urban Atmosphere and Its Deviation from Equilibrium Partitioning Using Random Forest Model.**

**Anonymous Referee #2**

*The manuscript by Wang et al. presents a well-executed study that integrates FIGAERO-CIMS measurements with a machine learning approach to understand the gas-particle partitioning of OOMs in urban environments. They identified key influencing factors such as relative humidity, liquid water content, and particle-phase composition. The authors showed how these factors affect the deviations from equilibrium partitioning, contributing significant knowledge to atmospheric aerosol science. I recommend publication after the authors address the following questions.*

**Response:**

We sincerely thank the reviewer for their valuable comments and suggestions, which will significantly improve the quality of our manuscript.

1. *It would be better to add more details about the sampling site's characteristics, such as location and distances to pollution sources (e.g., traffic or industrial areas), and discuss how these factors influence OOM partitioning.*

**Response:**

We appreciate the reviewer's valuable suggestion. In lines 72-80, we have added more detailed information about the characteristics of the sampling site:

"Hourly measurements of OOMs in both gas and particle phases was conducted during a winter campaign from December 5th, 2022, to January 8th, 2023, using an iodide-based FIGAERO-CIMS (Aerodyne Research Inc., USA) at a suburban site in Wuhan, a megacity in central China (114.6157°E, 30.4577°N). The site is located in the campus of China University of Geosciences, which is surrounded by residential and agricultural mixed area. The nearest urban center and industrial area are about 25 km west to the measurement site. Nearest highways and major roads lie about 2 km north and south of the site. The site is the only provincial supersite operated by local environmental authority for monitoring air quality in Wuhan and can thus be regarded as a receptor site influenced by wide ranges of emission sources from neighboring regions."

In lines 441-443, we have also expanded our discussion on the influence of wind speed and direction on OOM partitioning:

"As shown in Figure 6a, wind speed and direction rank relatively low in terms of feature importance for the six OOMs. This suggests that while wind direction and speed might

influence the source areas of OOMs, they have a minimal impact on the G/P ratios of OOMs."

2. *The authors use different $C^*$ parameterizations to estimate the equilibrium G/P ratios and compare them with the observed G/P ratios. However, these different $C^*$ parameterizations may introduce significant uncertainty. Please provide a more detailed discussion of the advantages and limitations of each parameterization and explain why these specific methods were chosen.*

**Response:**

We appreciate the reviewer's suggestion. Based on the existing literature, we selected the parameterization methods to estimate $C^*$ and compared their predictions. We have provided a more detailed discussion of the results from each method and explained the rationale behind our choice of these specific approaches in lines 283-306:

"Among all the methods, Mohr et al. (2019) predicts the steepest dependence of $\left(\frac{G}{P}\right)_{eq}$ on MW. Their $\left(\frac{G}{P}\right)_{eq}$ are higher than $\left(\frac{G}{P}\right)_{obs}$ for the OOMs with $n_C$ = 2-5 and lower than $\left(\frac{G}{P}\right)_{obs}$ for the OOMs with $n_C > 8$ (Figure 1b). It has been recognized by Kurtén et al. (2016) and subsequent publications that SIMPOL-derived parameterizations predict a too steep dependence of $C^*$ on Mw and oxygen content. Moreover, the parameterization of Mohr et al. (2019) likely produces $C^*$ of pure compounds. Without considering the effect of particle matrix, it may be unrealistic to predict G/P ratios using their $C^*$ parameterization. On the basis of thermal desorption temperature, Ren et al. (2022) predicts lower equilibrium G/P ratios than all other parameterizations and our observation. The weakness of Ren et al. (2022) is thermal desorption may result in the formation of decomposed fragments, which could be misidentified as OOM species. As a result, the $T_{max}$ of OOM formulas tends to be overestimated and the $C^*$ tends to be underestimated in their parameterization. Although Peräkylä et al. (2020) also predicted lower G/P ratios, their ratios are much closer to our observation than Ren et al. (2022). Among all the predictions, the prediction from Priestley et al. (2024) is most close to our observation. This is because their $C^*$ parameterization is based on the measured gas and particle-phase concentrations of OOMs in fresh or aged residential wood-burning emissions. Their predicted G/P ratio is thus inherently consistent with the observed G/P ratios in our study. This also highlights the risks of estimating volatility ($C^*$) using the partitioning method, which is based on measuring equilibrium gas- and particle-phase concentrations of OOMs. Two key issues arise: (1) OOMs may not achieve the assumed equilibrium state in real atmospheric or chamber conditions, introducing substantial uncertainty into calculations of $\left(\frac{G}{P}\right)_{eq}$; (2) The method fails for the compounds with extremely high or low volatility, as their gas- or particle-phase concentrations often fall below the detection limit of mass spectrometers. These limitations explain why the partitioning method typically reports a narrow volatility range (Voliotis et al., 2021; Chen et al., 2024)."

3. *The authors mentioned that the model is based on winter data and limited to specific OOM species. It is recommended that the authors clearly discuss the model's limitations, such as whether certain OOM types (e.g., highly volatile organics) were underrepresented, and how seasonal variations (e.g., summer heat or rainy season humidity) might affect model predictions. The authors should address these limitations and suggest future improvements, such as expanding the dataset to include different seasons or OOM species.*

**Response:**

We appreciate the reviewer's suggestion. We added the discussions about the limitations and future improvement direction in the end of the conclusion section.

"At last, the random forest models developed in this study have certain limitations. (1) Aerosol particle coating may serve as an inhibitory factor of gas/particle partitioning. However, the mixing state and morphology of aerosol particles were not considered in the model due to the challenges in quantifying these features with high resolution. (2) The OOMs with extremely high or low volatility might be underrepresented in this study, because their gas- or particle-phase concentrations often fall below the limit of quantification of FIGAERO-CIMS. (3) Isomers were not differentiated in the measurement of FIGAERO-CIMS in this study. The observed G/P ratio was contributed by isomers sharing the same chemical formula. The machine learning model built in this study did not account for the effect of isomerization on gas-particle distribution of OOMs. (4) The model was based solely on the data collected during the winter season and for specific groups of OOM species present in urban atmosphere. To enhance the robustness of the gas-to-particle partitioning model, future data collection under a broader range of atmospheric conditions is recommended."

4. *In the feature selection process, the authors identify essential features using the random forest model and explain them with SHAP analysis. It is recommended that the authors provide more statistical justification for the feature selection, such as significance tests or correlation analysis, to validate the importance of these features. Additionally, the SHAP interpretation could be enhanced by including quantitative analysis of how each feature's variation impacts the G/P ratio, not just the ranking of feature importance. Specifically, sensitivity curves for different features across ranges could visually show their contribution to the model output.*

**Response:**

We added a correlation analysis between the features and the observed G/P ratio of the six selected OOMs.

On lines 375-381, we added the following revision:

"All models show acceptable generalization ability ($R^2$ = 0.51-0.88). For all six OOMs, particle composition features dominate over meteorological and gaseous composition

features in predicting the G/P ratios (Figure 5). Particle composition features LWC, OC, $K^+$, $SO_4^{2-}$ and pH, as well as RH, consistently play important roles in influencing the G/P ratios of these species. This is roughly in line with the correlation analysis between the features and the observed G/P ratios of the selected 6 OOMs (Figure S5), which show that pH, RH, LWC, and $SO_4^{2-}$ exhibited strong positive or negative correlations with the G/P ratios."

In the supplementary materials, we add a new Figure S5.

[Figure]

**Figure S5.** The correlation between the observed G/P ratios of the 6 selected species and various features.

We had shown the sensitivity curves of SHAP values versus features across different feature ranges visually by Figure 7. Now we added quantitative interpretation of the SHAP values and sensitive ranges of features in Section 3.2.2. The blue front below highlights our changes:

[revised manuscript text omitted]

---

## Author Comment (AC3)

**A point-to-point response and relevant changes made in the revised manuscript**

**Ms. Ref. No.: egusphere-2025-229**

**Title: Investigating Influencing Factors of Gas-Particle Distribution of Oxygenated Organic Molecules in Urban Atmosphere and Its Deviation from Equilibrium Partitioning Using Random Forest Model.**

**Anonymous Referee #3**

*General comments*

*Wang et al. have conducted a measurement campaign using a FIGAERO-CIMS using Iodide to detect OOMs in Wuhan during the winter 2022-2023. The deployed instrument was able to record concentrations of both compounds in gas and particle phases. Using the data the authors developed several random forest models, attempting to predict the gas to particle partitioning of detected compounds. Models were also constructed to study the discrepancy between observed and modelled partitioning ratios. The significance of the features in each model were further analyzed in an attempt to elucidate physicochemical properties impacting the underlying processes.*

*The manuscript fits the scope of Atmospheric Chemistry and Physics, and presents valuable new knowledge. However, there are a number of concerns when it comes to the analysis presented. Therefore further clarifications are recommended before final publication.*

> **Response:**
>
> We sincerely thank the reviewer for their valuable comments and suggestions, which will significantly improve the quality of our manuscript.

*Specific comments*

*1. I suggest the authors provide some further explanation of why the selected target variable is chosen. This would help readers know what to expect from the analysis. What are the benefits of attempting to predict the gas to particle partitioning rather than e.g. absolute particle phase concentration?*

> **Response:**
>
> In this study, the gas-to-particle ratio is selected as the target variable because the phase distribution of OOMs is critical for understanding their volatility, atmospheric transformation pathways, and environmental impacts. Describing the benefits of predicting gas-particle partitioning is the core focus of the Introduction section.
>
> It presents greater challenges to predict absolute particle-phase concentrations of OOMs with machine learning models due to their strong dependence on diverse emission sources from

neighboring regions. We lack reliable features for quantifying the variable strengths of unknown sources and atmospheric aging processes during transport, which are key factors influencing the OOM concentrations.

In lines 223-230, we added:

"Third, single-species models were tailored to predict the gas/particle partitioning behaviors of these six individual OOMs under varying meteorological and gas-particle composition conditions. We also built random forest models to investigate how $\left(\frac{G}{P}\right)_{obs}$ of the six OOMs deviate from $\left(\frac{G}{P}\right)_{eq}$ under varying meteorological conditions and gas/particle compositions. In this study, we did not build random forest model to predict absolute gas or particle phase concentrations of OOMs, due to their strong dependences on diverse emission sources from neighboring regions. We lack reliable features for quantifying the variable strengths of unknown sources and atmospheric aging processes during transport, which are key factors influencing the OOM concentrations."

2. *Given the data first approach chosen, the quality of the dataset is important. Given that the datasets represents a time period of one month in winter, how representative is it of varying conditions? Are there several different meteorological conditions or weather patterns included in the analysed dataset, and what is their significance for the analysis? I encourage the authors to give a brief overview of the importance of varying conditions, and perhaps provide some time series of common meteorological parameters such as temperature and windspeed+direction in the supplementary as an overview for the reader.*

**Response:**

We appreciate the reviewer's suggestion. We added an overview of the observation period in lines 263-270:

"Despite the overall improvement in air quality in recent years, $PM_{2.5}$ episodes still occur frequently in December and January in most Chinese cities, contributing to the majority of $PM_{2.5}$ exceedance days of a year. During the winter observation period of this study, $PM_{2.5}$ concentrations ranged from 20 to 150 μg m$^{-3}$, spanning both clean and severe pollution conditions. Organic aerosol ($C_{OA} = C_{OC} \times 1.4$) comprised 10%–76% of $PM_{2.5}$, emerging as a critical bottleneck of eliminating $PM_{2.5}$ episodes. Time series of criteria pollutants and key meteorological parameters are presented in Figure S2. The data collected during the observation period herein is considered representative of winter $PM_{2.5}$ pollution characteristics in Wuhan."

We added Figure S2 in the supplementary materials:

[Figure]

**Figure S2.** Time series of partial features. (a) Wind direction and wind speed (reference vector: west 2 m s$^{-1}$). (b) PM$_{2.5}$ and organic carbon (OC) concentrations. (c) Ozone (O$_3$) and sulfur dioxide (SO$_2$) concentrations. (d) Ambient temperature and relative humidity (RH).

3. *Given the (it seems) limited scope of the dataset, how general are the conclusions of this work?*

**Response:**

As we stated above, t*he data* collected in this study is considered representative of winter PM$_{2.5}$ pollution characteristics in a megacity. Without more data from other environments like forest, rural or remote, we are unable to tell how general the conclusions of this work are. Like what we stated in the title of this manuscript, we focused only in the case of "urban atmosphere".

In the end of Conclusion Section, we stated the limitation of our dataset and recommended future direction:

"The model was based solely on the data collected during the winter season and for specific groups of OOM species present in urban atmosphere. To enhance the robustness of the gasto-particle partitioning model, future data collection under a broader range of atmospheric conditions is recommended."

*4. What is in general the certainty of the partitioning ratio measured? For example it is stated that 26.8% of particle mass is detected as fragments. Does this not bias the G/P ratio towards the gas phase for compounds that fragment? What about other losses in the FIGAERO system, or fragments that are not detected by the ionization scheme used?*

**Response:**

In line 121, we clarified "These fragments were excluded from the gas/particle partitioning analysis". So the fragment did not bias the G/P ratio of OOMs.

By comparing with the OC measured with the thermal-optical method, the OOMs measured with the FIGAERO-CIMS accounted for only $26 \pm 8\%$ of the total OA (OC $\times$ 1.4). So this study focused only on detectable OOMs by iodide ionization scheme, not all aerosol organic compounds.

In line 99-102, we add:

"According to our earlier investigation (Wang et al. 2024), the OOM measured with the FIGAERO-CIMS stands for only those polar and moderate-volatility organic species being desorbed below 200°C and accounted for only $26 \pm 8\%$ of the total OA (OC×1.4) measured with the thermal-optical method using the IMPROVE protocol."

*5. Also, the authors select the data based on peaks being relatively dominant on their masses, and with substantial concentration in the particle phase (lines 104-105). This excludes compounds with very small concentrations in either particle or gas phase. This, in turn, leads to a narrow range of G/P values, as compounds predominantly in either phase are filtered out. This is observed (Fig. 1), and should be commented on more.*

**Response:**

Yes, this was designed to obtain reliable concentrations and thus G/P ratios.

We have added more discussion in lines 114-117:

"In order to obtain reliable concentrations and thus G/P ratios, only those OOMs with a unit mass peak area ratio of > 20 % and a sample-to-blank ratio of > 2 were included for further analysis. This filtered out the OOMs with small concentrations in the atmosphere, as well as those extremely high or low volatility OOMs that are predominantly in only one phase."

In lines 501-503:

"The OOMs with extremely high or low volatility might be underrepresented in this study, because either gas- or particle-phase concentration of them often fall below the limit of quantification of FIGAERO-CIMS."

*6. On the comparison with previous $C^*$ parameterisations: Mohr et al parameterisation is still based on SIMPOL (through Tröstl et al), although with increased contribution from OOH groups. This is not clear in the text. Kurten et al (2016, 10.1021/acs.jpca.6b02196) and subsequent publications show that for HOM-type compounds, SIMPOL predicts a too steep dependence of $C^*$ on e.g. oxygen content and molar mass. Also, Peräkylä et al do not use particle-phase concentrations, and there is no assumption of equilibrium. In contrast, Priestley et al assume equilibrium.*

**Response:**

We appreciate the reviewer's comment, and we have revised the text as follows:

In lines 152-156, we made the following revision:

"Based on the saturation concentrations of HOM detected by Tröstl et al. (2016), Mohr et al. (2019) applied an updated version of a SIMPOL-type parameterization described by Donahue et al. (2011) to estimate $C^*$ from the numbers of carbon, oxygen, and nitrogen atoms of an organic species ($n_C$, $n_O$, and $n_N$), but emphasizing the increased importance of OOH groups."

In lines 160-167, we made the following revision:

"Peräkylä et al. (2020) parameterized the dependence of $C^*$ on $n_C$, $n_O$, $n_N$ and number of hydrogen atoms ($n_H$) by comparing steady-state gas-phase concentrations of α-pinene ozonolysis products with and without seed addition in a chamber. This parameterization predicts much smaller sensitivities of HOMs volatility to oxygen-containing functional groups than SIMPOL. The parameterization of Priestley et al. (2024) was based on measured gas and particle-phase concentrations, at an assumed equilibrium state, in residential wood-burning emissions. The $C^*$ of the products were obtained via Eq. (4) and a parameterization was obtained between $C^*$ and $n_C$, $n_O$, $n_N$ and $n_H$."

In lines 283-287, we add:

"Among all the methods, Mohr et al. (2019) predicts the steepest dependence of $\left(\frac{G}{P}\right)_{eq}$ on MW. Their $\left(\frac{G}{P}\right)_{eq}$ are higher than $\left(\frac{G}{P}\right)_{obs}$ for the OOMs with $n_C$ = 2-5 and lower than $\left(\frac{G}{P}\right)_{obs}$ for the OOMs with $n_C > 8$ (Figure 1b). It has been recognized by Kurtén et al. (2016) and subsequent publications that SIMPOL-derived parameterizations predict a too steep dependence of $C^*$ on MW and oxygen content."

7. *The authors compare the observed G/P with equilibrium G/P from other studies. When this comparison is presented to the reader, the reason behind the comparison is not clear, given that these are very different quantities. As the authors themselves state, OOMs rarely achieve equilibrium partitioning in the free atmosphere. I suggest the authors clearly motivate why the comparison is being made.*

**Response:**

Thank you for your suggestion.

In line 65-67, the end of Introduction Section, we clarified:

"By building data-driven machine learning models with the G/P ratio as the target variable, we explored the influencing factors of gas-particle distribution of OOMs and examined the factors that contribute to the deviations from equilibrium gas/particle partitioning."

In line 225-226, we added:

"We also built random forest models to investigate how $(\frac{G}{P})_{obs}$ of the six OOMs deviate from $(\frac{G}{P})_{eq}$ under varying meteorological conditions and gas/particle compositions."

8. *In Figure 1 the errorbars denote the range of observations, but systematic errors (such as those mentioned in comment 4) are not mentioned. Although these may not that relevant for the model, they may impact the absolute comparison presented here.*

**Response:**

Please see our response to Comment 4. The fragments were excluded from the gas/particle partitioning analysis. So the fragment issue did not bias the observed G/P ratio of OOMs.

The observed G/P ratio were calculated from equation 3 in the manuscript. The uncertainty comes only from the integrated signals of gas- and particle-phase OOMs during the measurement. Therefore, we believe the G/P ratios were not systematically biased.

$$(\frac{G}{P})_{obs} = \frac{C_g}{C_p} = \frac{signal_g \times t_p \times Q_p}{signal_p \times t_g \times Q_g}$$

9. My understanding is that the Ren et al. comparison is based on the parametrization derived from thermal desorption temperatures. It seems that the authors cloud also have derived some volatility estimate from their data since it was collected using the FIGAERO. Why is this not presented for further comparison?

**Response:**

Yes, we can also derive $C^*$ from $T_{max}$ obtained using our FIGAERO data; however, this merely replicates the work of Ren et al. The comparison of $C^*$ obtained from different methods has already been addressed by many other researchers (Chen et al., 2024; Stark et al., 2017).

Our manuscript focuses on building a machine learning model to explore influencing factors of G/P distribution of OOMs in the atmosphere and its deviation from equilibrium partitioning.

10. Do the authors believe they mostly observed OOMs close to equilibrium partitioning? Would this still be the case during summer, when the changes in precursors and oxidants are presumably faster?

**Response:**

In this study, the observed G/P ratios do not necessarily reflect a state close to equilibrium partitioning. We did not mention anywhere the observed G/P ratios are close to equilibrium partitioning. What we did is to compare $(\frac{G}{P})_{eq}$ and $(\frac{G}{P})_{obs}$ and to investigate the factors contributing to the discrepancy between them.

11. *On line 255 the authors state that they observed significant fluctuations in observed G/P diurnal variation. in the referenced Figure 2 the concentrations don't look like they vary very much relative to the mean, and the diurnal patterns look mostly random. I do not understand what the authors mean by this statement.*

**Response:**

We are sorry for the misunderstanding created by our wording. We rephrased the sentence in lines 313-319 as follows:

"In contrast, we observed different patterns of $(\frac{G}{P})_{obs}$ diurnal variations for the six OOM species during the campaign, as shown in Figure 2c-2h. This indicates that the extent of deviation of actual gas/particle partitioning from equilibrium state fluctuates randomly over time, driven by other unknown factors. In this study, we will first examine the influencing factors of gas-particle distribution of OOMs in urban atmosphere during the winter campaign (Section 3.2), followed by an investigation into the factors contributing to the discrepancies between observed and equilibrium G/P ratios (Section 3.3)."

12. *Since the diurnal variation of temperature and concentration of organic aerosol are highlighted as the only factors influencing the diurnal variation of equilibrium G/P I would like to see time series and/or diurnal plots of these parameters.*

**Response:**

We added a new Figure S2 in the supplementary materials to show time series of criteria

pollutants and key meteorological parameters, including temperature and OC.

13. *The authors claim that the importance of pH is due to enhanced partitioning of acidic OOMs. This argument relies on the assumption that the partitioning ratio is close to equilibrium for a significant fraction of the observations, and the observed G/P is mostly determined by factors shifting the equilibrium partitioning, which has not been shown in the manuscript. Could there be other reasons such as a common source for gas phase OOMs and more acidic particulate matter e.g. sulfuric acid?*

**Response:**

The partitioning always goes in a direction to reach a new equilibrium. But it does not necessarily mean the observed G/P ratios must be close to equilibrium partitioning G/P. We totally agree with you that the observed G/P is mostly determined by factors shifting (either facilitate or inhabit) the equilibrium partitioning. Under the influence of those factors, the partitioning ratio could be either close to or deviate from equilibrium. All we did here is that we found a decrease of the observed G/P ratio was associated with an increase in pH, on the basis of a large set of observation data and machine learning model. This makes sense because, obviously, the partitioning of acidic OOMs from gas to particles will be enhanced with elevated particle pH.

14. *Line 338 prohibited should probably be changed to inhibited, or another word.*

**Response:**

We have revised the text and changed "prohibited" to "inhibited."

15. *The authors hypothesize that elevated $O_3$ leads to depletion of OOMs in the gas phase. Does $O_3$ not also contribute to OOM formation? Can particle phase OOMs not react with $O_3$?*

**Response:**

We revised our statement to make it more precise. In lines 469-472, we have amended the text as follows:

"Since $O_3$ is not expected to change $\left(\frac{G}{P}\right)_{eq}$, the negative impact of $O_3$ on $\left(\frac{G}{P}\right)_{obs}/\left(\frac{G}{P}\right)_{eq}$ ratio could be explained by the fact that high $O_3$ concentrations are likely to deplete gas-phase OOMs at a faster rate than particle-phase OOMs, thereby reducing $\left(\frac{G}{P}\right)_{obs}$."

16. The models identified RH, LWC, O3 and temperature as influential factors driving the deviation between observed and equilibrium G/P. Are these truly influential, or do they serve as proxies for the diurnal pattern present mostly in the equilibrium G/P.

**Response:**

By itself, the machine learning model cannot uncover the fundamental mechanisms through which the factors influence G/P ratio. Apparently, temperature is a truly influential factor that changed the $\left(\frac{G}{P}\right)_{eq}$ and subsequently the discrepancy between observed and equilibrium G/P.

On the other hand, RH, LWC, $O_3$ are likely to serve as proxies of underlying oligomer formation, hydrolysis reaction or $O_3$ depletion reactions.

17. I suggest adding the term "Random forest" or "Machine learning" to the title since the paper mostly focuses on using these methods to study the partitioning. There are also few definitive conclusions about the influencing factors and processes, with the main conclusion being the importance of particle phase composition and processes.

**Response:**

Thank you for the suggestion. We have revised the title to:

"Investigating Influencing Factors of Gas-Particle Distribution of Oxygenated Organic Molecules in Urban Atmosphere and Its Deviation from Equilibrium Partitioning Using Random Forest Model"

18. Results from the same measurement campaign have already been published by Wang et al. (2024, 10.1021/acsestair.4c00076). This is completely fine, but it would be good to mention this clearly in the manuscript.

**Response:**

Thank you very much for your suggestion. We have clearly addressed this in the manuscript. In lines 118-121, we revised the text as follows:

"According to our earlier study on the same dataset using a K-means clustering method (Wang et al., 2024), on average, 25.1% of particle-bound OOM species number and 26.8% of OOM mass detected by the FIGAERO-CIMS could be attributed to thermal decomposition fragments (see Supplementary Materials Text S2)."

19. The authors use very many explanatory variables in their models. Many of these are correlated with, or even derived from, each other. Examples include the O/C ratio with oxidation state of carbon, and RH, sulfate and potassium concentrations with LWC and pH from ISORROPIA II. This leads to unreasonable conclusions, such as that O:C and oxidation state have an opposite effect on the G/P ratios. The authors should comment on problems with multicollinearity.

**Response:**

You raised an important issue.

We have found this issue when we observed the opposite effects of O:C and oxidation state, and DBE and H/C. This is due to the fact that these features are dependent variables as a

function of $n_C$, $n_H$, $n_N$ and $n_O$. To isolate the effect of oxidation and unsaturation-related features, we utilized the trained random forest model to predict G/P ratios of modified $C_{10}$ monocarboxylic acid with varying number of hydroxyl group and DBE (Figure 4). Other features in the model were fixed at average daytime or nighttime values observed during the campaign. These discussions are shown in line 357-366 in Section 3.2.1.

We have added a comment on multicollinearity issues in lines 194-201:

"This feature selection scheme guarantees a balanced representation of pertinent factors, while preserving the simplicity and predictive efficacy of the models. Unlike neural networks and other machine learning algorithms, the random forest model used in this study is an ensemble model made up of multiple decision trees. During training, each tree splits using a randomly chosen subset of features. Because each tree uses different feature subsets, this randomness in feature selection reduces the model's reliance on any single feature, making it less likely to be severely impacted by multicollinearity. To further ensure model stability, we also conducted five-fold cross-validation to confirm the robustness of the model."

20. Also, the supporting measurements (such as OC and aerosol composition) are poorly described in the methods.

**Response:**

We have added the following at lines 81-85 of the manuscript:

"We obtained valid data of 594 hours, during which meteorological parameters (e.g., relative humidity (RH) and temperature), particulate chemical components (e.g., organic carbon (OC) and sulfate ions ($SO_4^{2-}$)), and gaseous components (e.g., sulfur dioxide ($SO_2$) and ozone ($O_3$)) were routinely monitored. Detailed information about those routine measurement is shown in the supplementary materials (Text S1)."

In the supplementary materials, we have added:

"Text S1 Routine measurement of gaseous and particulate components

In this study, we measured the chemical composition of $PM_{2.5}$, including water-soluble ions (sulfate ($SO_4^{2-}$), nitrate ($NO_3^{-}$), ammonium ($NH_4^{+}$), chloride ($Cl^{-}$), and potassium ion ($K^{+}$)) over 594 hours using an Online Ion Chromatography Monitoring System (MARGA-1S, Metrohm AG, Switzerland) for the water-soluble ions. The system is designed to collect and analyze $PM_{2.5}$ in ambient air in real-time. Air samples are first passed through a cutter and sampling tubes into the instrument, where aerosol particles are captured and mixed with water vapor in a high-temperature steam generator. This process causes the particles to grow by absorption, after which they condense and are directed to the sample collection unit. The

collected samples are then separated and analyzed using ion chromatography for their water-soluble ion content, including both cations and anions.

The carbonaceous materials were analyzed using an OCEC Analyzer (RT-4, Sunset Laboratory Inc., USA), which employs a stepwise heating pyrolysis-oxidation method. The sample is first heated under helium (He) gas, causing the OC to volatilize and partially convert to pyrolyzed carbon (PC). The sample is then further heated in a helium/oxygen (He/O$_2$) mixture, where EC is oxidized and decomposed into gaseous oxidation products. All decomposition products flow through a carrier gas into an oxidation furnace, where the carbon products are converted to $CO_2$ and quantitatively detected using non-dispersive infrared (NDIR) methods. During this process, laser transmittance is used to monitor the OC/EC separation point, with OC volatilization and pyrolysis causing a decrease in transmittance intensity and EC oxidation leading to an increase. When the transmittance intensity returns to its initial level, the OC/EC separation point is defined, allowing for the precise determination of OC and EC content in the sample.

PM$_{2.5}$ concentrations were measured using an air particulate monitor (TH-2000PM, Wuhan Tianhong Technology Co., Ltd.), utilizing a dual-channel beta-ray method coupled with a dynamic compensation system. The air flow, set to 33.34 L/min, is first passed through a PM$_{10}$-cutting device to separate the larger particles. The flow is then evenly split into two streams: one stream directly measures PM$_{10}$, while the other is passed through a PM$_{2.5}$-cutting device for the measurement of PM$_{2.5}$. This approach reduces measurement errors and ensures accurate quantification of both PM$_{10}$ and PM$_{2.5}$ concentrations.

Hourly concentrations of nitrogen dioxide (NO$_2$), sulfur dioxide (SO$_2$), ozone (O$_3$), and ammonia (NH$_3$) were detected using gas analyzers (42i/43i/49i/17i, Thermo Fisher Scientific, USA). SO$_2$ was measured via pulsed fluorescence technology, in which SO$_2$ molecules absorb ultraviolet light at a specific wavelength and re-emit fluorescent light. The intensity of the emitted fluorescence is directly proportional to the SO$_2$ concentration. O$_3$ was quantified using ultraviolet photometry, as ozone absorbs ultraviolet light at a specific wavelength (254 nm). The O$_3$ concentration is determined by measuring the intensity of the absorbed light. For NH$_3$ measurement, the method involves its reaction with oxygen at high temperatures (750°C), converting NH$_3$ into nitrogen monoxide (NO). The NO concentration is directly proportional to the NH$_3$ concentration, which is then calculated accordingly. The concentration of NO$_2$ is measured using chemiluminescence: the sample first passes through a molybdenum catalyst at 325°C to convert NO2 into NO, which then reacts with O$_3$ generated by a silent discharge ozone generator, producing chemiluminescence detected by a photomultiplier tube (PMT). By measuring in NO and NO$_X$ modes—recording the NO concentration without catalysis and the total NO (including converted NO from NO$_2$) with catalysis—the NO$_2$ concentration is calculated as the difference between the two measurements.

Meteorological parameters, including relative humidity (RH), temperature (T), wind speed (WS), and wind direction (WD), were collected using an automatic weather station.

Photolysis frequencies of HONO ($J_{HONO}$) were measured with a PFS-100 photolysis spectrometer (Focused Photonics Inc., China). The spectrometer uses a quartz receiver head to collect solar radiation from various directions and transmits the light through optical quartz fibers to the spectrometer. The spectrometer then transmits the spectral data to an industrial computer, which calculates the photolysis flux. By integrating this flux with known absorption cross-sections and quantum yields, the photolysis rate constant is determined."

21. Add reference for Eq. (4)

**Response:**

We have added a reference for Eq. (4).:

"According to modified Raoult's Law, the saturation ratio of an organic species in gas phase (i.e. $\frac{C_g}{C^*}$) equals the mass fraction of the species in organic aerosol with mass concentration $C_{OA}$ ($i.e. \frac{C_p}{C_{OA}}$), under the assumptions of equilibrium absorptive partitioning of the species over an ideal organic solution and that the species has a molecular weight similar to that of the organic solution (Donahue et al., 2009; Epstein et al., 2010). The equilibrium G/P ratio $\left(\frac{G}{P}\right)_{eq}$ can thus be estimated from saturated mass concentration $C^*$ and mass concentration of organic aerosol $C_{OA}$ ($C_{OA} = C_{OC} \times 1.4$) using Eq. (4)

$$\left(\frac{G}{P}\right)_{eq} = \frac{C^*(T)}{C_{OA}} \qquad (4) \text{ "}$$

22. Data availability: I strongly recommend that the data of the study should be made openly available.

**Response:**

We uploaded our data to public data repository Zenodo and updated our **Data Availability Statement:**

[revised manuscript text omitted]

---

## Author Response (AR2)

**Response to the comments of the Editor and the Reviewers**

**Ms. Ref. No.: egusphere-2025-229**

**Title: Influencing Factors of Gas-Particle Distribution of Oxygenated Organics in Urban Atmosphere and Deviation from Equilibrium Partitioning: A Random Forest Model Study**

**Editor Comments**

1. *Thank you for revising your manuscript. Please provide a further revised manuscript addressing the minor technical points outlined by the final review, prior to publication.*

*Also, ACP prefers titles that highlight the scientific results/findings or implications of the study. Please adjust the title of your manuscript to adress this.*

Repones:

Thank you very much for your suggestion. We have revised the title as follows:

Influencing Factors of Gas-Particle Distribution of Oxygenated Organics in Urban Atmosphere and Deviation from Equilibrium Partitioning: A Random Forest Model Study.

**Referee Comments**

1. *In response to reviewer 3 comment 4 the authors write:*

*'In line 121, we clarified "These fragments were excluded from the gas/particle partitioning analysis". So the fragment did not bias the G/P ratio of OOMs.*

*By comparing with the OC measured with the thermal-optical method, the OOMs measured with the FIGAERO-CIMS accounted for only 26 ± 8% of the total OA (OC × 1.4). So this study focused only on detectable OOMs by iodide ionization scheme, not all aerosol organic compounds.'*

*It seems the authors did not quite understand the bias referred to in the comment. To clarify, the detection of thermal fragments implies the thermal decomposition of some compounds in the FIGAERO, presumably including the OOMs the authors investigated. By excluding the fragments, although they should be excluded, some of the signal originating from the OOMs of interest is excluded from the particle phase data. This will lead to an underestimation of the particle phase data, and therefore an overestimation of the detected G/P partitioning ratio. I*

*would like the authors to either explain how they were able to counteract this bias with their analysis, or clearly state that there is a slight bias towards lower particle phase concentrations in the data they present in the paper.*

Repones:

Thank you for pointing out the misunderstanding. Our analysis is valid for the OOMs that did not decompose at all. On the other hand, those OOMs that fully thermal-decomposed were naturally excluded from our analysis. For those OOMs undergoing partial thermal decomposition, to unknown extent, in the particle phase, our analysis might underestimate their particle-phase concentrations, thus biasing towards higher G/P ratios. We tend to believe that once an OOM is thermally unstable in the FIGAERO, it decomposes completely. However, we were unable to validate this hypothesis using the method in our analysis.

In lines 116-130, we have revised the text:

"Thermal desorption may cause OOM decomposition in the particle phase. According to our earlier study on the same dataset using a K-means clustering method (Wang et al., 2024), on average, 25.1% of particle-bound OOM species number and 26.8% of OOM mass detected by the FIGAERO-CIMS could be attributed to thermal decomposition fragments (Supplementary Materials Text S2). These fragments were excluded from the gas/particle partitioning analysis. The overlap of non-fragment particle-bound OOM species with those gas-phase OOM species resulted in 123 species, which were chosen as the target species for subsequent partitioning analysis. Based on our previous work (Figure S1) (Wang et al., 2024), these 123 OOM species were classified to 41 aromatic species (33.7%), 35 monoterpene-derived species (28.3%), 14 isoprene-derived species (11.4%), 11 aliphatic species (8.7%), 10 biomass burning tracers (8.1%), 3 sulfur-containing species (2.4%) and 9 other unknown species (7.3%). Notably, we cannot rule out the possibility that some of these 123 OOMs underwent partial thermal decomposition in the particle phase to an unknown extent. This could lead to an underestimate of their particle-phase concentrations and, in turn, bias the results toward higher G/P ratios."

*2. On line 153 the authors write:*

*'Based on the saturation concentrations of HOM detected by Tröstl et al. (2016)…'*

*I would like to note that those are modelled saturation concentrations, not measured, so the word detected is misleading. The study also assumes certains structures that have not been directly observed.*

Repones:

In lines 154-158, we have revised the text as follows:

Based on the saturation concentrations of HOMs modeled by Tröstl et al. (2016), Mohr et al. (2019) applied an updated version of SIMPOL-type parameterization described by Donahue et al. (2011) to estimate $C^*$ from the numbers of carbon, oxygen, and nitrogen atoms of an organic species ($n_C$, $n_O$, and $n_N$), but emphasizing the increased importance of OOH groups.

3. *On lines 469-472 the authors write:*

*'Since $O_3$ is not expected to change $(\frac{G}{P})_{eq}$, the negative impact of $O_3$ on $(\frac{G}{P})_{obs}/(\frac{G}{P})_{eq}$ ratio could be explained by the fact that high $O_3$ concentrations are likely to deplete gas-phase OOMs at a faster rate than particle-phase OOMs, thereby reducing $(\frac{G}{P})_{obs}$.'*

*By stating "by the fact that high $O_3$ ..." the authors imply this is established knowledge. If this is an established fact the authors should provide a source for this claim, otherwise it should be reworded to show that it is a hypothesis, or speculation, by the authors.*

Repones:

In lines 468-471, we add a citation Kaur Kohli et al., 2023 and revised the text as follows:

Since $O_3$ is not expected to change $(\frac{G}{P})_{eq}$, the negative impact of $O_3$ on $(\frac{G}{P})_{obs}/(\frac{G}{P})_{eq}$ ratio could be explained by the speculation (Kaur Kohli et al., 2023) that high $O_3$ concentrations are likely to deplete gas-phase OOMs at a faster rate than particle-phase OOMs, thereby reducing $(\frac{G}{P})_{obs}$.

**References**

Kaur Kohli, R., S., R. R., R., W. K., and and Davies, J. F.: Exploring the influence of particle phase in the ozonolysis of oleic and elaidic acid, Aerosol Sci. Technol., 58, 356-373, https://doi.org/10.1080/02786826.2023.2226183, 2023.